# α-Synuclein oligomers slow down action potential firing and enhance dopamine release by increasing Cav2.2 currents in midbrain dopaminergic neurons

Giulia Tomagra[1,2], Anthony Battaglia[1], Claudio Franchino[1], Sara Bonzano[3], Federico Picollo[2,4], Giuseppe Chiantia[1], Antonio de Iure[5,7], Paolo Calabresi[6], Barbara Picconi[5,7], Emilio Carbone[1], Silvia De Marchis[3], Andrea Marcantoni[1,2] and Valentina Carabelli[1,2]

[1] *Department of Drug and Science Technology, University of Torino, Turin, Italy*

[2] *NIS Interdepartmental Centre, University of Torino, Turin, Italy*

[3] *Department of Life Sciences and Systems Biology and "NICO" Neuroscience Institute Cavalieri Ottolenghi, University of Torino, Turin, Italy*

[4] *Department of Physics, University of Torino, Turin, Italy*

[5] *University San Raffaele, Roma, Italy*

[6] *Neurological Clinic, Fondazione Policlinico Universitario Agostino Gemelli IRCCS, Roma, Italy*

[7] *Experimental Neurophysiology Laboratory, IRCCS San Raffaele Roma, Roma, Italy*

Handling Editors: Katalin Toth & Samuel Young

The peer review history is available in the Supporting information section of this article (https://doi.org/10.1113/JP288914#support-information-section).

**Abstract figure legend** Left: the spontaneous quantal release of dopamine (DA) occurs at very low frequency in control conditions. Right: exogenous α-synuclein potentiates Cav2.2 currents and DA release but drastically reduces the spontaneous firing rate of substantia nigra DA neurons.

**Abstract** The spontaneous firing activity of substantia nigra (SN) dopaminergic (DA) neurons is finely tuned by the autocrine inhibition mediated by D2 DA autoreceptors (D2-ARs) that activate GIRK2 channels. Despite this regulatory mechanism, the vulnerability of SN DA neurons may nevertheless increase due to an altered spontaneous firing activity of DA neurons, Ca$^{2+}$ dishomeostasis, mitochondrial stress, high dendritic arborization, aggregation of α-synuclein (α-syn), α-syn mutations, reduced levels of calbindin protein, etc. Although the intraneuronal

G. Tomagra and A. Battaglia contributed equally.

*The Journal of Physiology*

accumulation and the spreading of misfolded α-syn is a hallmark of full-blown Parkinson's disease, the effects produced by α-syn aggregation on neuronal functionality at the early onset of neuro-degeneration are still of debate. We previously reported that α-syn oligomers in the extracellular medium drastically inhibit the firing rate of midbrain neurons and significantly impair burst generation and network synchronization. Here, by combining conventional electrophysiology and cutting-edge technology of micro-graphitized diamond micro-electrode arrays, we confirm that exogenous α-syn effectively slows down the firing rate of SN DA neurons, but it also selectively upregulates Cav2.2 (N-type) $Ca^{2+}$ currents and consequently $Ca^{2+}$-dependent DA release. Thus, our data uncover a novel regulatory mechanism in SN DA neurons and demonstrate that exogenous α-syn alters the interplay among $Ca^{2+}$ entry, spontaneous firing and DA release causing DA accumulation in the extracellular milieu and intracellular $Ca^{2+}$ overload. Both processes may represent a target for future investigations to better understand the initial phases of SN DA neuron degeneration.

(Received 19 March 2025; accepted after revision 20 February 2026; first published online 16 March 2026)

**Corresponding author** G. Tomagra: Department of Drug and Science Technology, University of Torino, Turin, Italy. Email: giulia.tomagra@unito.it

### Key points

- We combined conventional electrophysiology and micro-graphitized diamond multi-electrode arrays to investigate the effect of exogeneous α-synuclein on cultured midbrain dopaminergic neurons isolated from substantia nigra.
- α-Synuclein oligomers slow down the firing rate of dopaminergic neurons and up-regulate Cav2.2 (N-type) $Ca^{2+}$ currents.
- Raised Cav2.2 currents in turn increase the depolarization-evoked dopamine release and the frequency of quantal exocytotic events.
- Overall, this mechanism causes dopamine accumulation in the extracellular milieu and intra-cellular $Ca^{2+}$ overload.

## Introduction

Dopamine (DA) release from substantia nigra (SN) DA neurons occurs from axon terminals projecting to the striatum and from the somato-dendritic compartment. Released DA, in turn, exerts an autocrine inhibition on the firing activity of DA neurons (Lacey, 1993). This effect is turned on by the activation of dopamine D2 autoreceptors (D2-ARs) that activate inwardly rectifying potassium channels (GIRK2) (Anzalone et al., 2012; Dragicevic et al., 2014; Gantz et al., 2015; Lüscher & Slesinger, 2010) and inhibit N- and P/Q-type $Ca^{2+}$ channels (Cardozo & Bean, 1995). This autoregulatory mechanism is also involved in the limitation of additional DA release (Hikima et al., 2021), suggesting a strict coupling between cell firing and DA release.

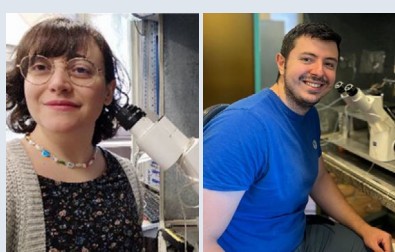

**Giulia Tomagra** received her Master Degree in Physics in 2016 from the University of Torino. In May 2021, she obtained a PhD in Neuroscience *cum laude* at the University of Torino under the super-vision of Professor Valentina Carabelli. The dissertation focused on monitoring the spontaneous activity and dopamine release from midbrain dopaminergic neurons using conventional electrophysiological techniques and diamond-based prototypes; focused on the effects of exogenous α-synuclein. She is currently a post-doctoral fellow at the University of Torino focused on the role of calcium channels in Parkinson's disease. **Anthony Battaglia** earned his master's degree in Cellular and Molecular Biology from the University of Turin in April 2022. In 2023, he received a fellowship from the Department of Neuroscience Rita Levi Montalcini, studying synaptic mechanisms in the somatosensory cortex related to the rare neurodevelopmental disorder CDD (CDKL5 deficiency disorder). At the end of 2023, he began his PhD in Neuroscience in the Department of Pharmaceutical Science and Technology under Professor Valentina Carabelli, focusing on Parkinson's disease, particularly how exogenous alpha-synuclein affects synaptic activity and neuronal excitability.

So far, quantal DA exocytosis has been detected and quantified using carbon fibres both in SN neurons of rat midbrain slices (Jaffe et al., 1998) and in cultured midbrain DA neurons (Pothos, 2002). More recently, midbrain DA neurons dissociated from SN were cultured on a novel micro-graphitized diamond multi-electrode array (µG-D-MEA) consisting of 16 recording microelectrodes that allowed multi-site detection of quantal exocytosis with high time resolution (Tomagra et al., 2019b). We demonstrated that amperometric spikes could be detected both during and in the absence of stimulation, proving that DA neurons exhibit sustained spontaneous activity and DA release during baseline activity. Despite several reports supporting the involvement of voltage-gated Cav channels in the regulation of somato-dendritic DA release, the identification of the Cav subtypes involved is still controversial (Chen et al., 2006). A predominant role of L- and T-type channels in sustaining somato-dendritic DA release has been demonstrated by means of amperometric recordings (Kim et al., 2008), whereas a role of P/Q-type $Ca^{2+}$ channels appears relevant in the presence of high $K^+$ concentrations, both in midbrain slices and in dissociated cells (Elverfors et al., 1997). In addition, there is also evidence that N-type channels contribute to sustaining somato-dendritic DA release in cultured midbrain neurons (Mendez et al., 2011).

Concerning the involvement of Cav channels in sustaining the spontaneous firing of SN DA neurons, several reports suggest a role of L-type (Cav1.2, Cav1.3) and P/Q-type (Cav2.1) $Ca^{2+}$ channels, whereas N-type channels (Cav2.2) do not seem specifically involved (Bean, 2007; Puopolo et al., 2007; Surmeier et al., 2012). In particular, Cav1.3 channels, which activate at more negative voltages with respect to Cav1.2, drive sub-threshold depolarizations and repetitive $Ca^{2+}$ transients during interspike intervals (Bean, 2007). These sub-threshold $Ca^{2+}$ fluctuations are responsible for supporting the rhythmic pacemaking activity (Dragicevic et al., 2014) and maintaining basal SN DA release (Surmeier & Schumacker, 2013). Thus, while the interplay among Cav channels and dopamine D2-ARs finely tunes the electrical activity of SN DA neurons and DA release, the imbalance of these mechanisms may lead to intracellular $Ca^{2+}$ overload, subsequent metabolic mitochondrial stress and neuron degeneration (Ford, 2014; Mosharov et al., 2009). These alterations probably represent the trigger for the progressive loss of SN DA neurons associated with the onset of Parkinson's disease (PD) (Liss & Roeper, 2010; Surmeier et al., 2012). It is of note that disruption of $Ca^{2+}$ homeostasis preferentially occurs in vulnerable SN DA neurons that exhibit low expression levels of $Ca^{2+}$ binding proteins, such as calbindin (Foehring et al., 2009; Lieberman et al., 2017), and are more prone to degeneration with respect to DA neurons in the neighbouring VTA region (Brichta & Greengard, 2014;

Guatteo et al., 2022; Ledonne & Mercuri, 2017). Among the causes that can lead to $Ca^{2+}$ dishomeostasis (for a detailed review see Martin et al., 2012), $\alpha$-synuclein ($\alpha$-syn) oligomers play a critical role. Concerning the action of $\alpha$-syn on midbrain DA neurons, we have previously shown that 48 h of incubation with 1 µM $\alpha$-syn drastically impairs spontaneous firing activity and network synchronism of DA neurons (Tomagra et al., 2023a). In addition, $\alpha$-syn applied extracellularly selectively increases $Ca^{2+}$ entry through Cav2.2 channels in rat cortical neurons and DA release when superfused on SN slices (Ronzitti et al., 2014). All this indicates the existence of an effective $\alpha$-syn-mediated pathway that regulates midbrain neuronal firing through Cav channel activation and DA release. Note that the effects of exogenous $\alpha$-syn on DA neurons is not limited only to Cav channels, as it can cause mitochondrial dysfunction (Wilkaniec et al., 2021), inhibit synaptic communication in cortical neurons (Hassink et al., 2018) and even regulate neuronal cholesterol efflux (Hsiao et al., 2017).

Here, we investigated the effect of exogenous $\alpha$-syn in midbrain DA neurons, with the aim of unravelling how $\alpha$-syn treatment affects Cav channels activity, DA release and spontaneous firing. We show that 48 h of incubation with 1 µM $\alpha$-syn drastically reduces spontaneous firing activity but increases the current amplitude of N-type (Cav2.2) channels and the corresponding DA release, measured through both µG-D-MEA amperometric recordings (Tomagra et al., 2019b) and depolarization-evoked capacitance changes. The paradoxical decreased firing frequency induced by $\alpha$-syn is rescued to normal values by blocking Cav2.2 (and Cav2.1) channels by $\omega$-MVIIC or by inhibiting D2-ARs with sulpiride (sulp), proving the involvement of a $Ca^{2+}$-driven D2-AR-mediated pathway that regulates neuronal firing in midbrain DA neurons. In conclusion, our data suggest that exogenous $\alpha$-syn potentiates DA release by selectively up-regulating Cav2.2 channels. The Cav2.2-mediated DA increase stimulates D2-ARs and the G$\beta\gamma$-coupled GIRK2 channels causing a consequent hyperpolarization of DA neurons and reduction of their spontaneous firing frequency. Thus, the increased DA release induced by $\alpha$-syn may represent a novel stress factor for SN DA neurons (Rice & Patel, 2015) that may down-grade action potential (AP) firing and nigro-striatal signals transmission.

## Materials and methods

### Ethical approval

Ethical approval was obtained for all experimental protocols from the University of Torino Animal Care and Use Committee, Torino, Italy. All experiments were conducted in accordance with the European Community's

Council Directive 2010/63/UE and approved by the Italian Ministry of Health and the Local Organism responsible for animal welfare at the University of Turin (Authorization 695/2020-PR). All animals were housed under a 12 h light/dark cycle in an environmentally controlled room with food and water available *ad libitum*. Every effort was made to minimize the number of animals used. For removal of tissues, animals were killed with exposure to a rising concentration of $CO_2$ and then rapidly killed by cervical dislocation.

### Primary cell culture of embryonic midbrain neurons

Mesencephalic DA neurons were obtained from dissection of SN (Matsushita et al., 2002; Sawamoto et al., 2001; Studer, 2001; Tomagra et al., 2023a). The ventral mesencephalon was dissected from embryonic day 16 (E16) C57BL/6 TH-GFP mice. TH-GFP mice were kept heterozygous via breeding TH-GFP mice with C57BL/6 mice (Bonzano et al., 2014, 2016).

Embryos were removed by caesarean section immediately after suppression by cervical dislocation. After being collected and extracted from the amniotic sac, these were decapitated and their heads placed in a Petri dish containing sterile digestion buffer (DB) [HBSS without $CaCl_2$ and $MgCl_2$ (thermofisher, life technologies, Paisley, UK) + 1% glucose 1 M + 1% BSA]. By means of a stereomicroscope, the heads were transferred to a siliconized dish containing DB and the excised brain. Next, the cerebral hemispheres, and the fore- and hindbrain regions were carefully removed and discarded. On the rostral side, a cut close to the thalamic region was performed to remove the superior colliculus. Once the ventral midbrain was isolated (SN), meninges were carefully removed using ultra-fine forceps. The dissected segments (without meninges) were then added, for 12 min at 37°C, into a digestion solution containing Papain (Worthington, PAPL; Lakewood, NJ, USA) and DNase (DNase I 15Ku, Sigma, D5025; St Louis, MO, USA). The solution was then replaced with Trypsin EDTA 0.25% (4 min at 37°C) and the fragments gently triturated. Finally, the medium solution [NBc high FBS: Neurobasal plus (Gibco) + 1% pen/strep (Sigma) + 1% L-glutamine + 10% fetal bovine serum (FBS, low IgG) + 2% B27 plus (Gibco)] was added and cells centrifuged at 1200 r.p.m. for 4 min. Once the supernatant was discarded, cells were plated at final densities of 800 cells/mm$^2$ on plastic dishes. Cultured neurons were used between 11 and 14 DIV (days *in vitro*).

Dishes were coated with poly-L-lysine (0.1 mg/ml) as substrate adhesion. Cells were incubated at 37°C in a 5% $CO_2$ atmosphere, with Neuro Basal Medium containing 1% pen-strep, 1% ultra-glutamine, 2% B-27 plus and 2.5% FBSd (pH 7.4).

To control glia proliferation, 5 mM 5-fluoro-2-deoxyuridine (FdU) from Sigma was added into dishes at 4 DIV. To preserve DA neurons, 20 ng/ml recombinant human brain-derived neurotrophic factor (BDNF, Sigma) was added into dishes at 4 DIV.

All experiments using drugs were performed by adding the drugs to the culture medium under static conditions. α-Synuclein (1 μM) was added to the culture medium for 48 h in an incubator at 37°C and 5% $CO_2$ (Tomagra et al., 2023a).

All experiments were performed at room temperature (RT), with the culture medium or solutions warmed to 37°C prior to the experiment.

### Patch-clamp recordings

Macroscopic whole-cell currents and APs were recorded using an EPC 10 USB HEKA amplifier and Patchmaster software (HEKA Elektronik GmbH, Lambrecht/Pfalz, Germany) following the procedures previously described (Baldelli et al., 2005; Gavello et al., 2018; Marcantoni et al., 2014).

Traces were sampled at 10 kHz and filtered using a lowpass Bessel filter set at 1 kHz. A subset of experiments was performed using a higher sampling rate (50 kHz) for a more precise evaluation of the parameter d$V$/d$t$ in the phase-plane plot. Borosilicate glass pipettes (Kimble Chase Life Science, Vineland, NJ, USA) with a resistance of 7–8 MΩ were used. Uncompensated capacitive currents were reduced by subtracting the averaged currents in response to P/4 hyperpolarizing pulses. Off-line data analysis was performed with pClamp 10.0 software for current clamp recordings. $Ca^{2+}$ currents were evoked by applying a single depolarization step (50 ms duration), from a holding of –70 to 0 mV. Fast capacitive transients due to the depolarizing pulse were minimized online by the patch-clamp analogue compensation. Series resistance was compensated for by 80% and monitored during the experiment.

Depolarization-evoked exocytosis was measured as membrane capacitance increases by applying a sinusoidal wave function on the holding potential (±15 mV amplitude, 1 kHz). Fast capacitive transients due to depolarizing pulses were minimized online by the patch-clamp analogue compensation.

### Amperometric recordings

Amperometric recordings were performed while maintaining the cells in the culture medium (see above), which contains 2 mM $CaCl_2$. Amperometric recordings were performed by means of μG-D-MEA (4 × 4 channel geometry) and dedicated electronics. The whole electronic chain was inserted into a Faraday cage to minimize

noise. The diamond chip was directly plugged in to the front-end electronics connected to a data acquisition unit (National Instruments USB-6216; Austin, TX, USA). The diamond-microarray circuit was grounded using a reference Ag/AgCl electrode, which was immersed in the extracellular medium solution (Marcantoni et al., 2023). Amperometry was performed by holding the 16 electrodes at a constant potential of +800 mV relative to the Ag/AgCl reference electrode (Tomagra et al., 2019a). The amplified signals were filtered at 10 kHz with 4th order Bessel low-pass filters and were subsequently acquired at a sampling rate of 25 kHz per channel. We used data acquisition control software that we have developed in LabView (Tomagra et al., 2019b).

## Solutions

For current clamp experiments the pipette internal solution contained (mM): 135 gluconic acid (potassium salt: K-gluconate), 10 Hepes, 0.5 EGTA, 2 $MgCl_2$, 5 NaCl, 2 ATP-Tris and 0.4 Tris-GTP. The extracellular solution (Tyrode solution) contained (mM): 2 $CaCl_2$, 10 Hepes, 130 NaCl, 4 KCl, 2 $MgCl_2$ and 10 glucose adjusted to pH 7.4.

For voltage-clamp recordings, the pipette internal solution contained (mM): 90 CsCl, 20 TEA-Cl, 10 EGTA, 10 glucose, 1 $MgCl_2$, 4 ATP, 0.5 GTP and 15 phosphocreatine adjusted to pH 7.4. For recordings of capacitance changes, the pipette internal solution contained (mM): 95 CsCl, 25 TEA-Cl, 0.1 EGTA, 10 glucose, 1 $MgCl_2$, 4 ATP, 0.5 GTP and 15 phosphocreatine adjusted to pH 7.4. The extracellular solution for voltage-clamp recordings contained (mM): 135 TEA, 2 $CaCl_2$, 2 $MgCl_2$, 10 Hepes and 10 glucose adjusted to pH 7.4, complemented with 300 nM TTX, 3 μM isradipine, 100 nM SNX-482 and 3.2 μM $\omega$-conotoxin GVIa to block, respectively, voltage-dependent, L-type $Ca^{2+}$ channels, R-type $Ca^{2+}$ channels and N-type $Ca^{2+}$ channels.

For capacitance changes experiments, the external solution contained (mM): 4 TEA-Cl, 118 NaCl, 2 $MgCl_2$, 10 Hepes, 10 glucose, 4 KCl and 10 $CaCl_2$ (pH 7.4 with NaOH). $\omega$-Conotoxin MVIIC (1 μM 10 min of incubation) was added and maintained in the extracellular solution to block N,P/Q-type $Ca^{2+}$ channels.

For both current- and voltage-clamp experiments, APV (2-Aminophosphonovaleric acid) (50 μM), 6,7-dinitroquinoxaline-2,3-dione (DNQX) (20 μM) and picrotoxin (100 μM) were added to the extracellular solution.

## Diamond multielectrode arrays

The μG-D-MEAs are fabricated from high-quality chemical vapour-deposited diamond substrates. The diamond crystals are classified as IIa with a low content of nitrogen (10 ppm) and boron (1 ppm) impurities, have a size of $4.5 \times 4.5 \times 0.5$ mm$^3$ and the two opposite faces are optically polished. The graphitic microelectrodes, as described previously in detail (Carabelli et al., 2017; Olivero et al., 2010; Picollo et al., 2010, 2015), are fabricated by ion beam lithography (1.8 MeV He ions delivered at a fluence of $1 \times 10^{17}$ cm$^{-2}$). These energies allow the formation of highly damaged regions embedded in the surrounding diamond matrix a few micrometres deep. The geometry of the implanted regions in the *x–y* plane is obtained using a 100 μm thick metal stencil mask that collimates the ion beam, while the emersion of the electrodes to the surface is achieved by focused ion beam (FIB) lithography. After the lithographic steps, the diamonds are subjected to high-temperature thermal annealing (950°C in high vacuum for 2 h), which promotes the transformation of the damaged region into highly conductive graphite. Microchannel end-points are created in correspondence of the central area of the sensors for interfacing with plated cells and in the periphery of the crystal for electrical connection with the front-end electronics. The electrodes have a length of 950–1200 mm, width of ∼20 μm and thickness of ∼250 nm, while the active area of the electrode has a size of ∼100 μm$^2$.

## Immunofluorescence

Cells plated on a glass coverslip were fixed with 4% PFA (pH 7.4) for 15–20 min and then stored in PBS containing 0.02% of sodium azide (NaN$_3$; 71 289, Sigma–Aldrich) at 4°C until staining. Cells were then incubated in blocking solution [0.01 M PBS, pH 7.4, 0.1% Triton X-100 (T8787, Sigma–Aldrich) and 10% normal donkey serum (NDS; S30-M, Sigma-Aldrich)] for 45 min at RT to permeabilize membranes and block non-specific epitopes. They were then incubated overnight at 4°C with primary antibodies anti-GFP (polyclonal Ab raised in chicken; 1:1000; GFP-1020, AvesLab) and anti-$\beta$-Tubulin III/Tuj1 (monoclonal Ab raised in mouse; 1:1000; t8660, Sigma–Aldrich) diluted in 0.01 M PBS (pH 7.4), 0.0025% Triton X-100 and 1% NDS. Following three washes in PBS, cells were incubated for 1 h in the dark at RT with secondary antibodies (AlexaFluor488-conjugated donkey anti-chicken; 1:400; 703-545-155, Jackson ImmunoResearch, West Grove, PA, USA; AlexaFluor647-conjugated donkey anti-mouse; 1:600; 715-605-151, Jackson ImmunoResearch) in 0.01 M PBS (pH 7.4). After washing in 0.01 M PBS (pH 7.4), cells were incubated with 4,6-diamidino-2-phenylindole (DAPI; 1 μg/ml; D9542, Sigma–Aldrich) for 10 min at RT to counterstain DNA. After three washes in 0.01 M PBS (pH 7.4), coverslips were air dried and mounted onto gelatine-coated slides using antifade mounting medium Mowiol (4-88 reagent; 475904, Calbiochem).

## Image acquisition and cell counting

Images were acquired using a Stellaris 5 confocal laser scanning microscope (Leica Microsystems, Wetzlar, Germany).

Confocal multi-stacked images were captured at 0.5 µm optical steps using a 40× oil immersion lens objective (HC PL APO CS2 40×/1.30 NA) with an additional zoom (1.5×), pinhole at 1.00 AU and with a resolution of 8 bit, 512 × 512 pixels and 200 Hz scan speed (1 voxel size: 0.379 × 0.379 × 0.346 µm; *xyz*). Images were captured randomly in a tile-like manner. Images were then analysed using Fiji (https://fiji.sc/).

DAPI segmentation to estimate nuclei was performed by a custom macro as follows: (i) a maximum *z*-projection was applied to obtain 2D images; (ii) the images from the DAPI channel were smoothed, filtered [(i) Gaussian Blur, sigma = 1; (ii) Median, radius = 4] and converted into binary mask by pixel intensity thresholding (with Default algorithm) to separate the objects from the background; and (iii) a watershed algorithm was applied to separate touching objects. Finally, the analyse particles tool (size = 15–infinity) was used to count cell nuclei. Neurons positive (+) for the pan neuronal marker Tuj1 (Tuj1+) and/or GFP under control of the TH promoter (GFP+; i.e. DAergic neurons) were manually counted by the Cell counter plug in. Results are expressed as a percentage of DAergic TH-GFP+ neurons on: (i) the total number of cultured cells (i.e. TH-GFP±/DAPI) and (ii) the subset of Tuj1+ neurons (i.e. TH-GFP±/Tuj1±).

## Statistics

Data are indicated in the text and figures as mean ± SD (Statistical summary). Data that were not normally distributed (Shapiro–Wilk and Pearson normality test) were analysed by means of the non-parametric Mann–Whitney test or Kruskal–Wallis ANOVA (KW-ANOVA, Origin Pro software).

The analysis of current-clamp and voltage-clamp recordings was performed using Pclamp10 software (Molecular Devices, Sunnyvale, CA, USA). Amperometric recordings were analysed using the APE software (Tomagra et al., 2023b). For comparison, data were validated via the 'Quanta Analysis' routine (Mosharov & Sulzer, 2005) in Igor Pro 5.00 data analysis software (WaveMetrics, Inc., Lake Oswego, OR, USA).

## Results

### α-Syn treatment does not affect the viability of cultured DA neurons

Using MEA recordings, we have previously shown that addition of 1 µM α-syn to the extracellular medium (48 h)

causes the formation of oligomers that have a dramatic effect on both the firing discharge of extracellularly recorded APs and neuronal network synchronization in cultured midbrain DA neurons (Tomagra et al., 2023a). Here, we further investigated the effect of α-syn oligomers selectively focusing on SN DA neurons identified through TH-GFP fluorescence staining. Extracellular α-syn (1 µM) was added to the culture medium at 9 DIV and analyses were performed 48 h after the incubation time (11 DIV) to allow the formation of oligomeric aggregates, as previously described (Ronzitti et al., 2014; Tomagra et al., 2023a).

To reveal the effects of α-syn treatments at the neuronal level, we assessed possible changes in the viability of neurons by immunofluorescence, quantifying the fraction of Tuj1+ neurons expressing the pan neuronal marker Tuj1 over the total number of nuclei and found no changes following α-syn treatment compared to controls (percenatge of Tuj1 on DAPI; ctrl: 6.30 ± 0.97; α-syn: 6.10 ± 0.93; Mann–Whitney test, $P = 0.342$). Next, by focusing on the subset of TH-GFP+ DA neurons, we did not find any difference in the number of the two groups when data were normalized both to the total number of nuclei (TH-GFP±/DAPI) and of Tuj1+ neurons (Fig. 1*A* and *B*). These findings indicate that 48 h of incubation with α-syn does not affect the viability of cultured midbrain neurons, including DA neurons.

### α-Syn oligomers reduce the spontaneous firing discharge of midbrain DA neurons

As previously reported using MEA recordings, α-syn oligomers drastically reduce the firing discharge in midbrain networks (Tomagra et al., 2023a). Here we focused on the action of α-syn oligomers on spontaneous neuronal firing, selectively focusing on TH-GFP-identified midbrain DA neurons. This effect was tested in current-clamp conditions, at 11 DIV after 48 h of preincubation with 1 µM α-syn (Ronzitti et al., 2014; Tomagra et al., 2023a).

Representative traces of spontaneously firing DA neurons are shown in Fig. 2*A* and *B*, respectively for untreated (ctrl) and α-syn-treated neurons. In control conditions, spontaneous AP firing occurred with a mean firing discharge of 4.6 ± 3.6 Hz ($n = 38$ cells), while the frequency decreased to 1.9 ± 2.1 Hz after α-syn exposure ($n = 50$ cells, $P < 0.001$, KW-ANOVA, Fig. 2*D*). Spontaneous activity can be recorded for minutes without significant alterations, though the analysis of frequency values and AP parameters was standardized and evaluated during the first 10 s of recording. We found that the reduced firing frequency was accompanied by an increased mean interevent interval (0.43 ± 0.42 s in control, $n = 37$ and 0.86 ± 0.68 s with α-syn, $n = 40$; $P = 0.001$, Mann–Whitney test, Fig. 2*E*).

After comparing the spontaneous firing in the absence and after α-syn incubation, we measured the effects of α-syn on evoked AP firing, by applying 10 s current injections of increasing amplitude, from 0 to +60 pA, with steps of 10 pA (Fig. 2*F*). As expected, both control and α-syn-treated DA neurons displayed progressively increased mean firing rates (Guatteo et al., 2017). However, the rate of increase of firing frequencies was significantly lower for control neurons with respect to α-syn-treated neurons. Linear fits respectively gave slopes of 0.13 ± 0.15 Hz/pA for control (untreated neurons, black line) and 0.30 ± 0.15 Hz/pA for α-syn-treated neurons (red line; *P* = 0.0038, *F* test). These data collectively indicate that spontaneous firing activity is inhibited by α-syn, whereas the firing rate is potentiated if stimulated by high input currents. In Fig. 2*C*, representative traces of evoked AP firing are shown at increasing current pulse

injections (from 0 to +60 pA). Finally, we measured the input resistance ($R_m$) for control and α-syn-treated neurons. $R_m$ was calculated as the reciprocal of slope conductance using linear regression of the steady-state voltage responses elicited by three current steps of increasing amplitude (−80, −100 and −120 pA) and 100 ms duration (Fig. 2*G*). We found that α-syn did not significantly alter the membrane resistance ($R_m$) (mean values were 0.47 ± 0.11 GΩ in control, *n* = 27 and 0.42 ± 0.0.11 GΩ after α-syn, *n* = 33; *P* = 0.375, Mann–Whitney test).

## α-Syn accelerates the AP upstroke and increases AP repolarization

To determine how α-syn alters the AP waveform, we compared the AP parameters generated by control (untreated) and α-syn-treated DA neurons, during their spontaneous activity (Fig. 3*A*). AP parameters were evaluated over 10 s. As shown in Fig. 3*C–G*, α-syn significantly reduced the AP half-width and the time to peak ($T_p$) while increased the AP after-hyperpolarization (AHP). In detail, we found that AP half-width decreased from 3.7 ± 1.18 ms (control, *n* = 37) to 3.2 ± 1.34 ms (α-syn, *n* = 41, *P* = 0.04, Mann–Whitney test). $T_p$ was reduced from 6.0 ± 0.4 ms (control, *n* = 37) to 5.8 ± 0.3 ms (α-syn, *n* = 41, *P* = 0.001, Mann–Whitney test), whereas AHP increased from −44.3 ± 5 mV (control, *n* = 37) to −46.7 ± 4.42 mV (α-syn, *n* = 41, *P* = 0.009, *t* test).

In addition, a phase-plane plot analysis of APs (Fig. 3*B*) uncovered a net increase in the maximum time derivative of voltage ($dV/dt_{max}$) induced by α-syn (red trace). During the AP up-stroke, $dV/dt_{max}$ increased from 107 ± 48 mV/ms in controls (*n* = 46) to 159 ± 44 mV/ms with α-syn (*n* = 30, *P* = 0.0001, one-way ANOVA, Fig. 3*F*), suggesting that α-syn causes a marked increase of the inward $Na^+$ and $Cav^{2+}$ currents sustaining the rapid rising of AP (Tomagra et al., 2019*b*). In contrast, no significant changes were found on the mean AP over-shoot (Fig. 3*G*). Mean values were 38.1 ± 10.3 mV in control (*n* = 37) and 41.8 ± 10.7 mV with α-syn (*n* = 41, *P* = 0.211, Mann–Whitney test). Concerning the AP threshold, we observed that it was significantly lowered after α-syn incubation (from −39.4 ± 3.8 mV in the control condition to −41.9 ± 4.5 mV with α-syn, *P* = 0.038, Fig. 3*H*).

It of note that the changes of the AP waveform detected in the presence of α-syn are similar to those induced by L-DOPA on the same neurons, that is increased $dV/dt_{max}$ and AHP (Guatteo et al., 2013; Tomagra et al., 2019*b*).

In a subset of experiments, current signals were sampled at 50 kHz and filtered at 10 kHz to better quantify the $dV/dt$ values in the phase-plane plot. Results are shown in Fig. 4. Even at a higher sampling

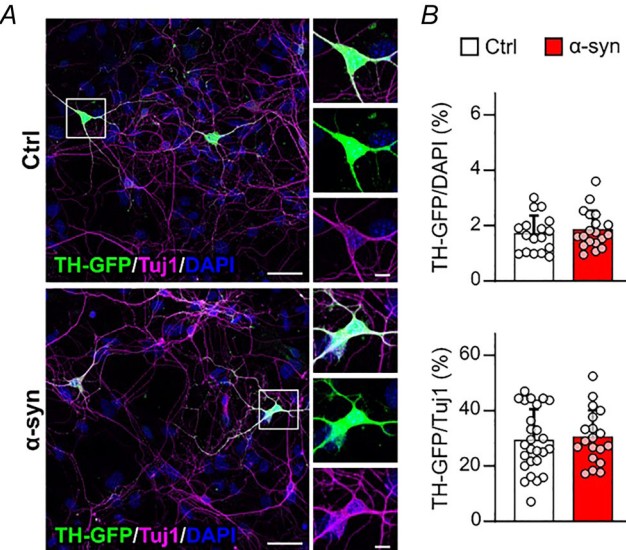

**Figure 1. DA neuronal viability is unaffected by α-syn treatment**
*A*, representative confocal images of midbrain DA neurons at 9 DIV immunostained for green fluorescent protein (GFP; green) and the pan neuronal marker Tuj1 (magenta) in control medium (Ctrl) and after exposure to α-synuclein (α-syn) oligomers for 48 h. Nuclei are counterstained with DAPI. Two examples of TH-GFP/Tuj1 double-positive neurons are shown in the inset. *B*, quantification of TH-GFP+ midbrain DA neurons among DAPI+ nuclei (top) and Tuj1+ neurons (bottom) in Ctrl *versus* α-syn-treated cultures. All percentages were obtained from at least five regions of interest (ROI) per replicate and derived from three biological replicates per condition (Ctrl: *n* = 17,834 DAPI nuclei, *n* = 1264 Tuj1+ neurons; α-syn: *n* = 25,173 DAPI nuclei, *n* = 1514 Tuj1+ neurons). Data are presented as mean ± SD and each dot represents an ROI. Mann–Whitney tests: *P* = 0.5569 (top), *P* = 0.6125 (bottom). Scale bars in *A* = 50 μm (low magnification) and 10 μm (high magnification). Results of cell counting are presented as mean ± SD, and they are derived from at least five images per coverslip and derived from three replicates (i.e. coverslips) per condition. DA, dopaminergic; DAPI, 4,6-diamidino-2-phenylindole.

rate, α-syn significantly increased $dV/dt_{max}$ (from $75.34 \pm 17.76$ mV/ms in ctrl to $143.40 \pm 40.22$ mV/ms in α-syn, $P = 0.0011$). In these cells, a kink was present in 55% of the cells and absent in the remaining 45%, regardless of the sampling rate (Spratt et al., 2021).

### Exogenous α-syn potentiates DA release: capacitance change measurements

We next investigated what causes the reduced AP firing rate induced by α-syn and the acceleration of the AP upstroke, as detected in the phase-plane plot analysis. As suggested by Ronzitti et al. (2014), in rat primary cortical neurons and in striatal brain slices, α-syn selectively activates Cav2.2 channels and potentiates DA release by increasing $Ca^{2+}$ entry.

We tested whether, also in our cell model, the addition of exogenous α-syn could potentiate the $Ca^{2+}$-dependent DA release. In this case, the increased DA release could be responsible for the autocrine activation of D2-ARs, and consequently for the slowing down of spontaneous AP firing activity through the activation of GIRK2 channels (Dragicevic et al., 2014).

To demonstrate this, we initially measured the depolarization-evoked secretion. The bath solution contained 10 mM $[Ca^{2+}]$ to maximize the secretory response (Moser & Neher, 1997). A mixture of isradipine (3 μM) and SNX-482 (100 nM) was added in the extracellular solution to block Cav1 and Cav2.3 channels,

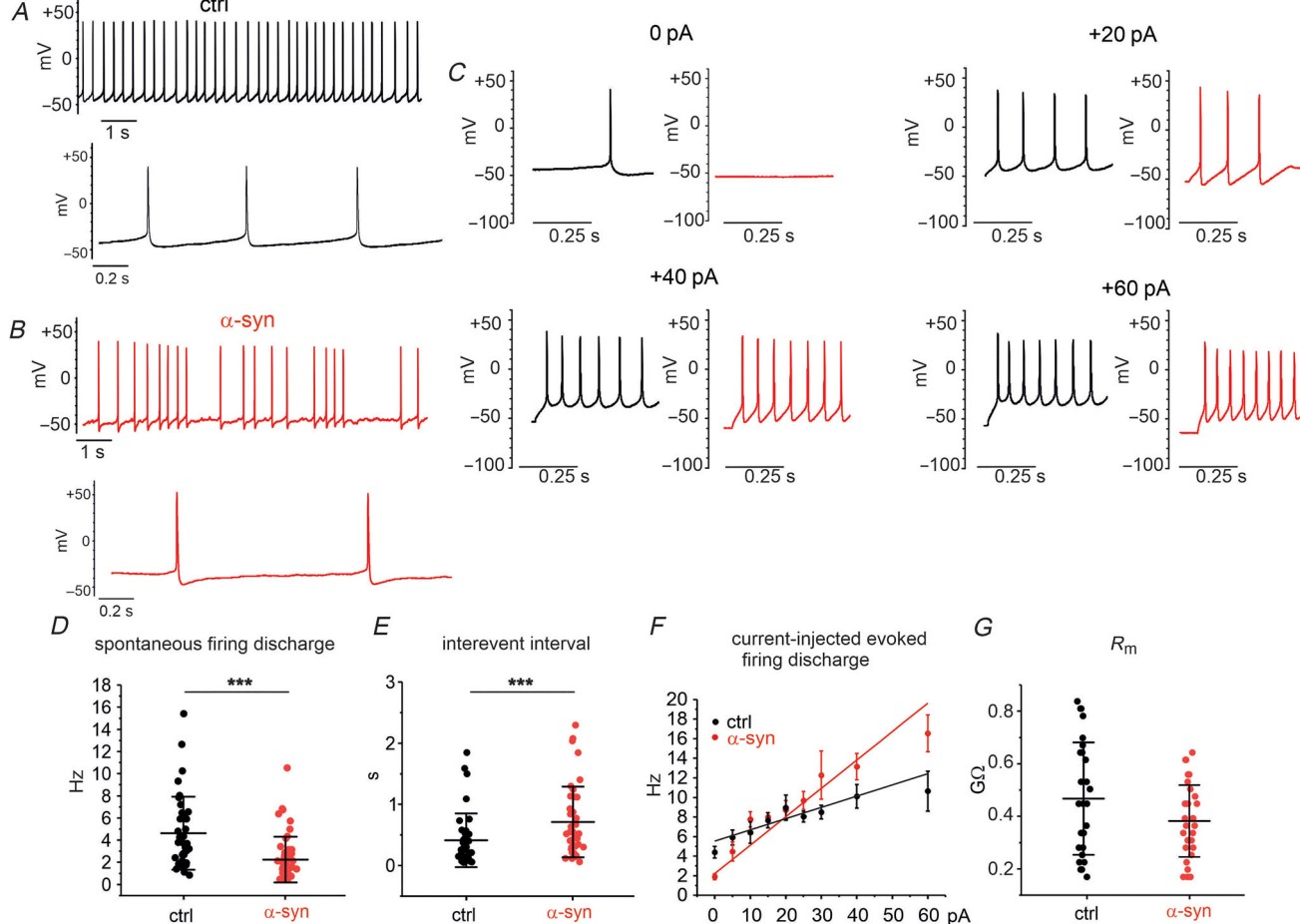

**Figure 2. Effects of α-syn on the firing activity of DA neurons**
*A*, representative traces of spontaneously firing DA neurons under control conditions and after α-syn exposure. *B*, insets: APs shown at higher magnification. *C*, representative traces of evoked AP firing under control conditions (black traces) and after α-syn exposure (red traces). *D*, scatter plot of spontaneous firing discharge values in control condition (ctrl, black dots) and after α-syn treatment (red dots; $P = 0.001$, KW-ANOVA). *E*, scatter plot of interevent interval values ($P = 0.001$, Mann–Whitney test). *F*, mean firing frequency values plotted *versus* current injection pulses (from 0 to 60 pA steps). Linear fit gave a $0.13 \pm 0.03$ Hz/pA slope for untreated neurons (ctrl, continuous black line) and $0.30 \pm 0.03$ Hz/pA for α-syn-treated neurons (red line). *G*, scatter plot of membrane input resistance values ($R_m$) ($P = 0.375$, Mann–Whitney test). α-syn, α-synuclein; ANOVA, analysis of variance; AP, action potential; DA, dopaminergic; KW-ANOVA, Kruskal–Wallis ANOVA.

respectively (Benkert et al., 2019). TTX was not added to prevent slowdown of Na$^+$ channel gating kinetics (Carabelli et al., 2003; Horrigan & Bookman, 1994). The quantity of charge, $Q$, was calculated as the time integral of the inward current. Given the presence of an early inward Na$^+$ current (since TTX was not used), the limits for the current integration were fixed 3–4 ms after the beginning of the pulse once 80% of the Na1 currents were decayed. Under these conditions, depolarization-evoked DA secretion from DA neurons is mainly mediated by Cav2.1 and Cav2.2 channels. DA neurons were stimulated by means of square pulses from −70 mV ($V_h$) to 0 mV for 100 ms; membrane capacitance increases ($\Delta C$) induced by DA secretion were estimated after the transient depolarization (Fig. 5$A$). By comparing control and $\alpha$-syn-treated neurons (1 μM, 48 h incubation), we found that mean $\Delta C$ increased from 26.5 ± 12.06 fF in control

($n = 11$) to 88.9 ± 59.32 fF for $\alpha$-syn -treated cells ($n = 15$) ($P = 0.001$, $t$ test; Fig. 5$B$), confirming that $\alpha$-syn significantly potentiates DA release. Furthermore, we found that the quantity of charge increased from 6.4 ± 6 to 22.1 ± 8 pC/pF ($P < 0.001$, $t$ test), confirming that, when Cav1 and Cav2.3 are blocked, $\alpha$-syn significantly upregulates the remaining Cav channels (Cav2.1 and Cav2.2).

As an additional confirmation, when we measured the depolarization-evoked secretion in the presence of $\omega$-conotoxin MVIIC (1 μM), to block Cav2.1 and Cav2.2 channels (Fig. 5$D$–$F$), we found that the secretory responses induced by Cav1 and Cav2.3 were not modified by $\alpha$-syn. No significant changes could be detected either for $\Delta C$ (38.8 ± 24.2 fF in control, $n = 8$ and 43.5 ± 31.5 fF in $\alpha$-syn-treated cells, $n = 8$; $P = 0.738$, $t$ test) or for the quantity of charge (pC/pF) (22.8 ± 10.26 pC/pF in control,

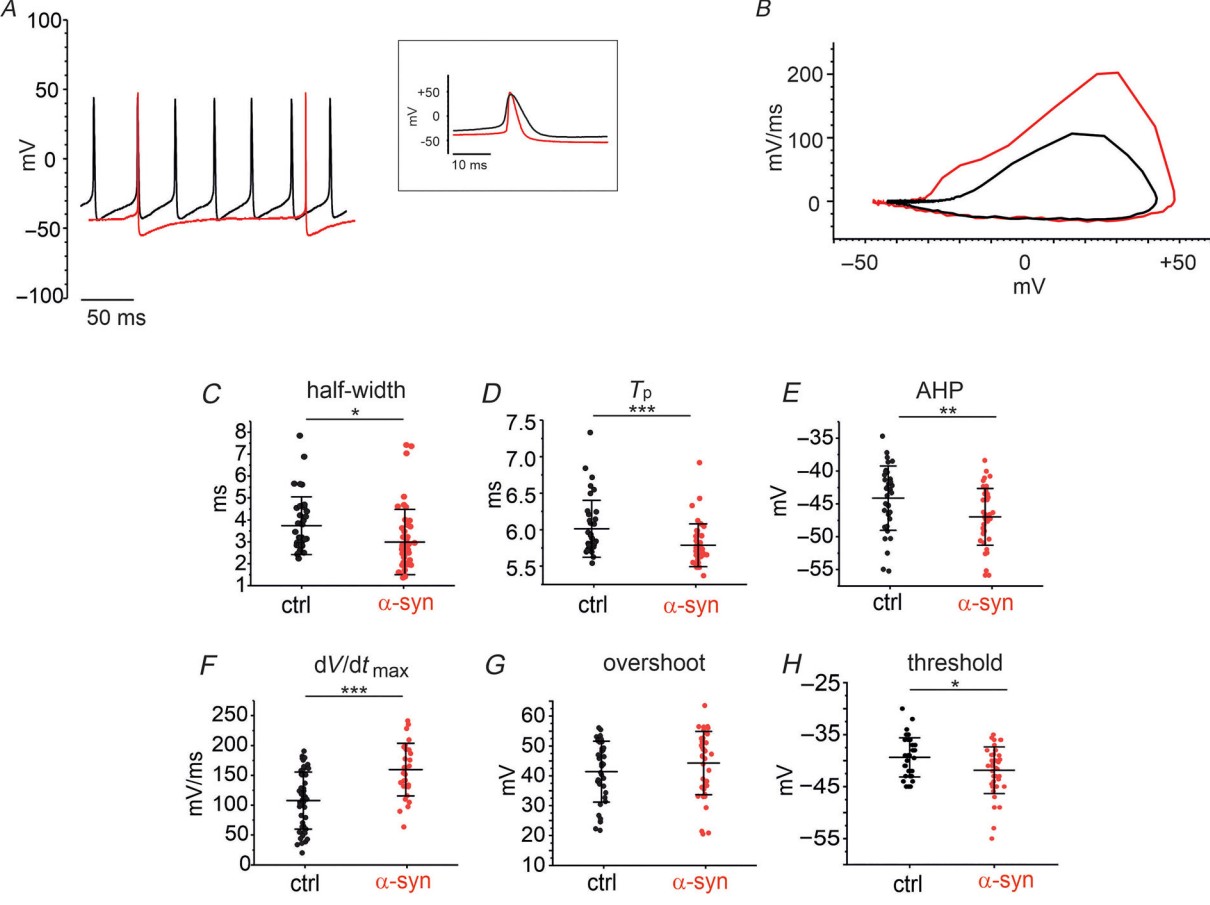

**Figure 3. $\alpha$-Syn modifies the AP waveform of DA neurons**
*A*, representative AP traces and relative single APs shown at higher magnification in the inset (control, black trace; $\alpha$-syn, red trace). *B*, phase-plane plot of AP obtained from the traces shown in *A*, inset. *C–G*, scatter plots of AP parameters. Control data are represented as black points; $\alpha$-syn as red points. *C*, scatter plot of AP half-width values ($P = 0.04$, Mann–Whitney test). *D*, scatter plot of AP time to peak ($T_p$, $P < 0.001$, Mann–Whitney test). *E*, scatter plot of AP after-hyperpolarization (AHP) values ($P = 0.009$, $t$ test). *F*, scatter plot of the AP maximum time derivative of voltage (d$V$/d$t_{max}$) values ($P < 0.0001$, one-way ANOVA). *G*, scatter plot of AP overshoot values ($P = 0.211$, Mann–Whitney test). *H*, scatter plot of AP threshold values ($P = 0.038$, Mann–Whitney test). $\alpha$-syn, $\alpha$-synuclein; ANOVA, analysis of variance; AP, action potential; DA, dopaminergic.

$n = 8$ and $23.5 \pm 11.46$ pC/pF in α-syn-treated cells, $n = 8$; $P = 0.893$, $t$ test). These data suggest that the potentiating effect of α-syn on Cav channels and secretion occurs limitedly to Cav2.1 and/or Cav2.2 channels. Since these experiments were carried out using 10 mM extracellular $[Ca^{2+}]$, we avoided to use the more selective Cav2.2 antagonist ω-GVIA because of the reduced blocking potency of the toxin in bath solution with high divalent cation concentrations (Boland et al., 1994; Gong et al., 2007). Overall, these experiments demonstrate that α-syn treatment is efficient in potentiating $Ca^{2+}$-dependent secretion when Cav2.1 and/or Cav2.2 channels are available.

### Exogenous α-syn potentiates DA release: μG-D-MEA amperometric measurements

Since membrane capacitance changes provide information only concerning the overall secretory response, in order to understand whether the increased secretory response could be due to an increased amplitude, frequency or altered time course of quantal exocytotic events, we performed amperometric recordings using μG-D-MEAs (Fig. 6) (Jaffe et al., 1998; Picollo et al., 2020; Tomagra et al., 2019a, 2019b).

This diamond-based multiarray prototype, patterned with 16 micro-graphitized DA sensing electrodes, allowed us to measure, in real time, quantal exocytotic events produced by several cultured DA neurons on the same chip.

We thus performed amperometric recordings to monitor DA release under control, after α-syn treatment (48 h of incubation), and after α-syn plus the Cav2.1/Cav2.2 blocker ω-MVIIC (1 μM). Representative amperometric recordings performed simultaneously from four electrodes (ch1, ch2, ch6 and ch11) are shown in Fig. 6B. Note that signals are acquired from the same electrodes initially under control conditions (black traces), after 48 h exposure to 1 μM α-syn (red traces) and after addition of 1 μM ω-MVIIC (blue traces). As shown in Fig. 6D, α-syn incubation increased nearly 10-fold the frequency of amperometric spikes (from $0.2 \pm 0.1$ to $1.9 \pm 0.9$ Hz, $n = 7$, $P = 0.003$, KW-ANOVA) in all four recording electrodes, confirming the increased rate of exocytosis following α-syn treatment. Interestingly, the addition of ω-MVIIC in α-syn-treated neurons reduced the frequency of amperometric spikes to $0.07 \pm 0.04$ Hz ($n = 5$, $P = 0.009$). This value is comparable to the one detected in control conditions ($P = 0.004$, KW-ANOVA), as expected if the potentiation of secretion is due to Cav2.2 channels.

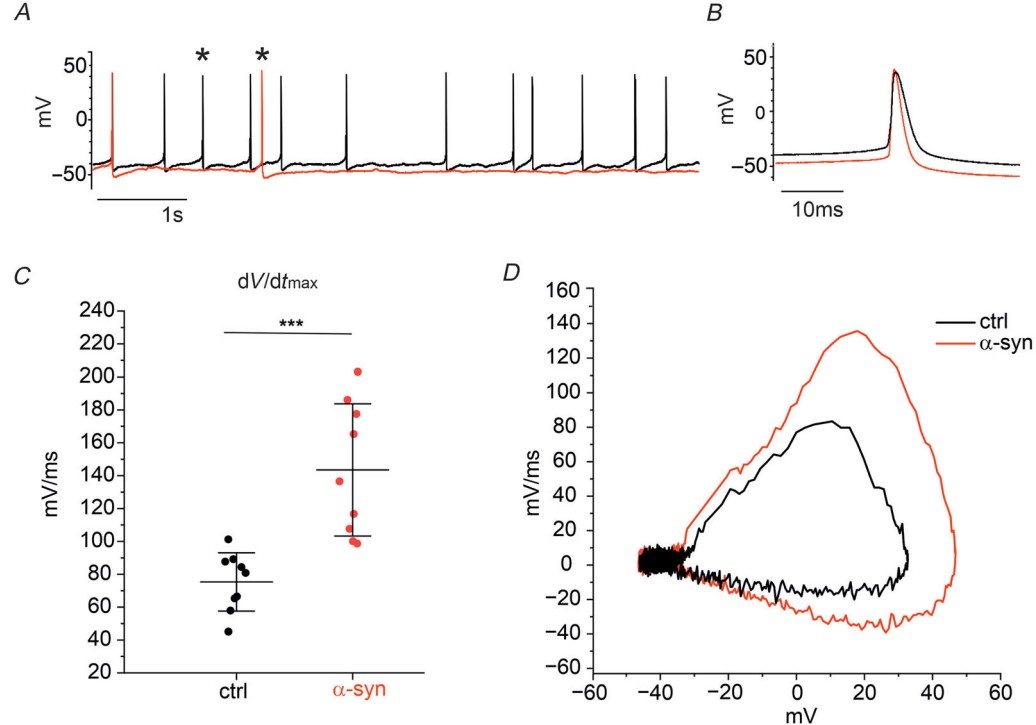

**Figure 4. Current signals sampled at 50 kHz**

*A*, representative AP traces, sampled at 50 kHz: control (black trace) and α-syn (red trace). *B*, comparison of single AP waveforms at higher time magnification: control (black trace), α-syn (red trace). *C*, scatter plot of the AP maximum time derivative of voltage ($dV/dt_{max}$) values ($P < 0.0001$, one-way ANOVA). *D*, phase-plane plot of AP obtained from the traces shown in *A*. α-syn, α-synuclein; ANOVA, analysis of variance; AP, action potential.

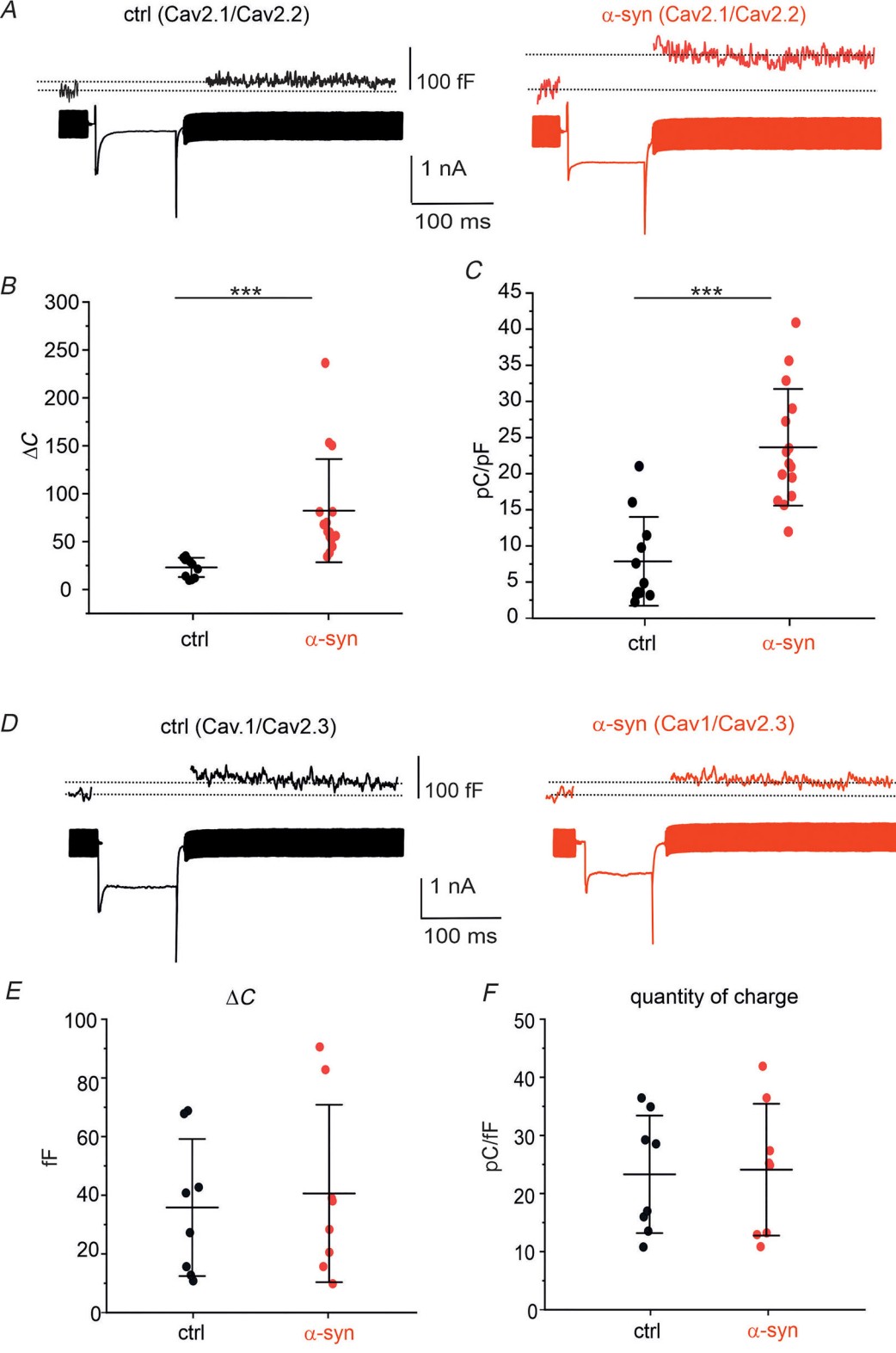

**Figure 5. Depolarization-evoked Ca²⁺-dependent secretion measured in untreated and α-syn-treated neurons**

*A*, depolarization-evoked $Ca^{2+}$ currents recorded at 0 mV in the presence of isradipine (3 μM) and SNX-482 (100 nM) to evaluate the contribution of Cav2.1 and Cav2.2 channels, in control (black) and after α-syn (red). At the top, the corresponding depolarization-evoked membrane capacitance increases (ΔC) are shown. *B*, scatter

plot of membrane capacitance increase values ($\Delta C$). Control values are significantly different from α-syn values ($P = 0.001$, $t$ test). C, scatter plot of quantity of charge values, normalized to cell capacitance (pC/pF). Control values are significantly different from α-syn ($P < 0.001$, $t$ test). D, depolarization-evoked secretion induced by Cav1 + Cav2.3 $Ca^{2+}$ channels. ω-Conotoxin MVIIC (1 μM) was added to the extracellular solution. Recordings were obtained from untreated neurons (ctrl, black) and after incubation with α-syn (red). E, scatter plot of $\Delta C$ values. Data obtained from untreated and α-syn-treated neurons are not significantly different ($P = 0.738$, $t$ test). F, scatter plot of quantity of charge values normalized to cell capacitance (pC/pF). Data obtained from untreated and α-syn-treated neurons are not significantly different ($P = 0.893$, $t$ test).

Besides monitoring the frequency of exocytotic events, the following amperometric spike parameters were evaluated (Fig. 6E–G): $I_{max}$ (maximum of the amperometric current generated by the oxidized molecules), $Q$ (quantity of charge, calculated as the integral of the amperometric current over time, indicating the amount of oxidizable DA reaching the electrode) and

$t_{1/2}$ (half width, the time at which the spike amplitude reaches 50% of its maximum value). We found that, after α-syn, $I_{max}$ significantly increased from $14.5 \pm 7.5$ pA (control, $n = 106$) to $23.5 \pm 16.8$ pA (α-syn, $n = 2993$, $P < 0.001$, KW-ANOVA), while this effect was reversed by ω-MVIIC (α-syn + ω-MVIIC, $n = 42$, $10.5 \pm 5.4$ pA, $P < 0.001$, KW-ANOVA). No variation was detected

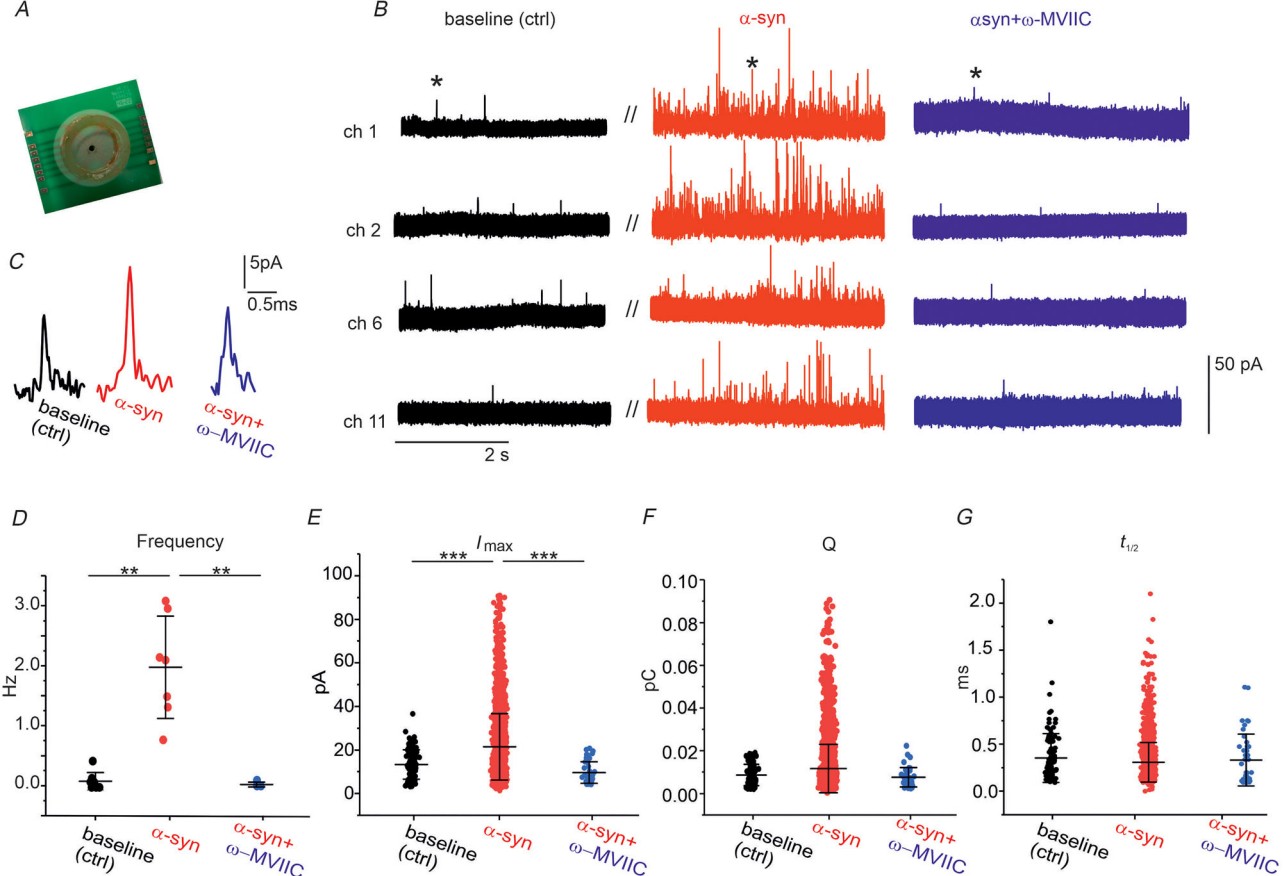

**Figure 6. α-Syn increases the frequency of quantal exocytotic events**
*A*, image of 4 × 4 channel μG-D-MEA. *B*, four representative amperometric traces are shown in control condition (black), after α-syn incubation (red) and after α-syn plus ω-MVIIC (blue). *C*, amperometric spikes, indicated by asterisks in *B*, are shown at higher magnification. *D*, scatter plot of amperometric spike frequency values. Statistically significant differences were evaluated by KW-ANOVA ($P = 0.002$, ctrl *vs.* α-syn-treated neurons; $P = 0.007$, α-syn *vs.* α-syn + ω-MVIIC). *E–G*, spike parameter values are compared among control condition (black), after α-syn incubation (red) and α-syn + ω-MVIIC (blue). *E*, scatter plot of maximum amperometric spike ($I_{max}$) values. Statistically significant differences were evaluated by KW-ANOVA ($P < 0.001$ ctrl *vs.* α-syn, $P < 0.001$ α-syn *vs.* α-syn + ω-MVIIC). *F*, scatter plot of quantity of charge ($Q$) values. G, scatter plot of half-width values ($t_{1/2}$). α-syn, α-synuclein; μG-D-MEA, micro-graphitized diamond multi-electrode array; ANOVA, analysis of variance; KW-ANOVA, Kruskal–Wallis ANOVA.

for the quantity of charge ($Q$) in the three conditions (control, $\alpha$-syn, $\alpha$-syn + $\omega$-MVIIC). Mean $Q$ values were 7.5 $\pm$ 5 fC in control, 11.1 $\pm$ 10 fC after $\alpha$-syn incubation ($P = 0.245$, KW-ANOVA) and 7.0 $\pm$ 4 fC after $\alpha$-syn + $\omega$-MVIIC ($P = 0.091$, KW-ANOVA). Regarding the amperometric spike half-width ($t_{1/2}$), this parameter was not significantly modified by $\alpha$-syn and by the combination of $\alpha$-syn + $\omega$-MVIIC. Mean $t_{1/2}$ values were 0.33 $\pm$ 0.23 ms (control), 0.29 $\pm$ 0.18 ms ($\alpha$-syn, $P = 0.634$, KW-ANOVA test) and 0.32 $\pm$ 0.23 ms ($\alpha$-syn + $\omega$-MVIIC, $P = 0.595$, KW-ANOVA test). Amperometric spike parameters detected in control conditions are in good agreement with those previously reported (Tomagra et al., 2019b).

## $\alpha$-Syn selectively increases Cav2.2 current density in mouse DA midbrain neurons

On the basis of the above findings, we aimed to separately test the effect of $\alpha$-syn on Cav1, Cav2.1 and Cav2.2 channels in midbrain DA neurons. In bath solutions containing isradipine (3 $\mu$M), SNX-482 (100 nM) and $\omega$-agatoxin IVA (2 $\mu$M) to block Cav1, Cav2.3 and Cav2.1 channels, we measured the effect of $\alpha$-syn on spared Cav2.2 channels (Antunes et al., 2023; Chi et al., 2009; Mueller et al., 2023). We found that Cav2.2 currents recorded during 100 ms square pulses to 0 mV ($V_h = -70$ mV) were drastically increased by $\alpha$-syn (Fig. 7*B* and *C*). The mean peak amplitude (normalized to the cell capacitance) increased from 21.1 $\pm$ 13.6 pA/pF ($n = 14$, control) to 78.7 $\pm$ 41.5 pA/pF ($n = 20$, $\alpha$-syn, $P = 0.044$, Mann–Whitney test).

Interestingly, when in addition to Cav1 and Cav2.3 channel blockers, we also inhibited Cav2.2 channels with 3.2 $\mu$M $\omega$-conotoxin GVIA, the remaining Cav2.1 current appeared clearly insensitive to $\alpha$-syn incubation (Fig. 7*D* and *E*). The mean Cav2.1 current amplitude was 67.0 $\pm$ 20.9 pA/pF in control ($n = 14$) and 67.5 $\pm$ 31.4 pA/pF with $\alpha$-syn ($n = 18$; $P = 0.809$, *t* test).

Finally, we tested the effect of $\alpha$-syn on Cav1 channels, after blocking Ca$_v$2.1, Ca$_v$2.2 and Ca$_v$2.3 using $\omega$-MVIIC (1 $\mu$M) and SNX-482 (100 nM). As shown in Fig. 7*F* and *G*, the Ca$^{2+}$ current amplitude mediated by Cav1 channels remained unaltered (52.2 $\pm$ 25.72 pA/pF in control, $n = 26$ *vs.* 45.5 $\pm$ 27.15 pA/pF with $\alpha$-syn, $n = 18$; $P = 0.210$, Mann–Whitney test), suggesting that exogenous $\alpha$-syn does not affect Cav1 channels.

In the absence of Cav channel antagonists, we could not find any difference in the total current amplitude between control and $\alpha$-syn (67.0 $\pm$ 21.0 pA/pF in control, $n = 14$ *vs.* 62.7 $\pm$ 24.6 pA/pF with $\alpha$-syn, $n = 17$; $P = 0.59$, Mann–Whitney test) (Fig. 7*H* and *I*). We then concluded that $\alpha$-syn increased DA secretion through a selective upregulation of Ca$_v$2.2 currents.

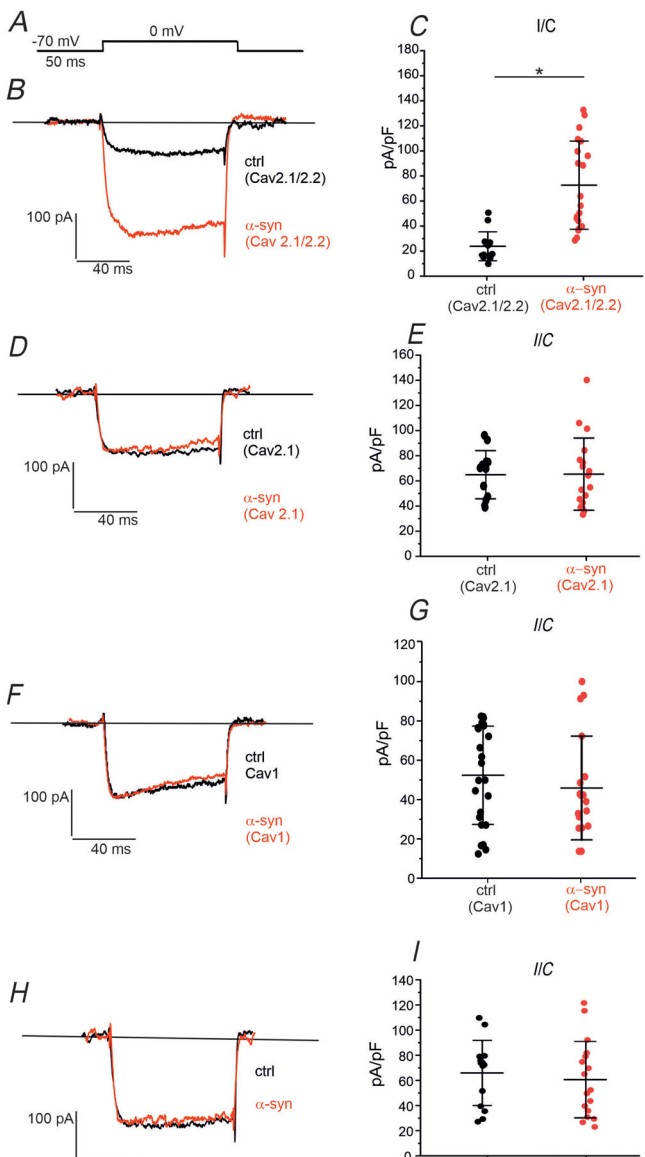

**Figure 7. $\alpha$-Syn effects on Cav channels in midbrain DA neurons**
*A*, the stimulation protocol consists of depolarizing steps to 0 mV (100 ms duration), from a holding potential of –70 mV. *B*, representative Cav2.2 current traces measured from untreated neurons (ctrl, black trace) and after $\alpha$-syn treatment (red trace). *C*, scatter plot of Cav2.2 peak current amplitude (normalized to the cell capacitance). Statistically significant difference is indicated by the horizontal line. *P* values are specified in the text. *D*, representative Cav2.1 current traces for untreated neurons (ctrl, black trace) and after $\alpha$-syn treatment (red trace). *E*, scatter plot of Cav2.1 peak current amplitude (normalized to the cell capacitance). *F*, representative Cav1 current traces for untreated neurons (ctrl, black trace) and after $\alpha$-syn treatment (red trace). *G*, scatter plot of Cav1 peak current amplitude (normalized to the cell capacitance). *H*, representative total calcium current traces for untreated neurons (black trace) and after $\alpha$-syn treatment (red trace). *I*, scatter plot of total calcium currents (normalized to the cell capacitance). $\alpha$-syn, $\alpha$-synuclein; DA, dopaminergic.

## ω-MVIIC or the D2R antagonist sulpiride rescue the normal AP firing slowed down by α-syn

After assessing that α-syn upregulates Cav2.2 channels and DA release, we tested whether these mechanisms could potentiate the autocrine D2-mediated loop and consequently slow down the spontaneous firing activity of midbrain DA neurons. By measuring the spontaneous firing discharge either in the presence of ω-MVIIC (1 μM, to maintain the same $Ca^{2+}$ channel blocker that we used for the $\Delta C$ recordings) or the D2-AR antagonist sulp (10 μM), we found that neither drug significantly altered the spontaneous firing frequency of control (α-syn untreated) neurons (Fig. 8C): mean frequencies were respectively 3.6 ± 2.9 Hz for ω-MVIIC ($n = 33$, $P = 0.273$, KW-ANOVA) and 4.2 ± 2 Hz for sulp ($n = 23$, $P = 0.688$). This is in good agreement with the observation that Cav2.2 channels contribute to the AP upstroke, but not to the AP onset (Gantz et al., 2015).

By contrast, in α-syn-treated neurons, ω-MVIIC completely reversed the reduction of the firing discharge induced by α-syn. Mean values were: 4.9 ± 3.4 Hz (α-syn + ω-MVIIC, $n = 37$) *versus* 1.9 ± 1.4 Hz, α-syn, $n = 50$ ($P < 0.0001$, KW-ANOVA) (Fig. 8A). Similar effects were induced by the D2R antagonist sulp, which increased the mean firing rate of α-syn−treated neurons to 4.6 ± 3.4 Hz ($n = 23$ $P < 0.002$, KW-ANOVA, with respect to α-syn−treated neurons). In summary, the decreased rate of firing discharge induced by α-syn is completely reversed by the addition of either MVIIC or sulp, confirming the involvement of $Ca_v2.2$ channels and D2-receptors.

Addition of sulp or ω-MVIIC to α-syn-treated neurons also reversed the effect of α-syn on the AP waveform. For example, the maximum rise slope of the AP, $dV/dt_{max}$, measured in α-syn + sulp or α-syn + ω-MVIIC neurons, was significantly different from the one of α-syn-treated neurons and comparable to the one obtained under control conditions. Mean $dV/dt_{max}$ values were 87 ± 49 mV/ms (α-syn + sulp, $n = 18$, $P < 0.001$ *versus* α-syn, one-way ANOVA) and 100 ± 41 mV/ms (α-syn + MVIIC, $n = 15$, $P < 0.001$, *vs.* α-syn, one-way ANOVA). Similarly, mean AHP was −42.7 ± 4.5 mV (α-syn + sulp, $n = 19$, $P = 0.012$, one-way ANOVA with respect to α-syn) and −43.8 ± 4.9 mV (α-syn + MVIIC, $n = 24$, $P = 0.025$, one-way ANOVA with respect to α-syn). Mean values of $T_p$ were 6.1 ± 0.3 ms (α-syn + sulp, $n = 19$, $P < 0.001$, one-way ANOVA with respect to α-syn) and 6.0 ± 0.3 ms (α-syn + ω-MVIIC, $n = 24$, $P = 0.038$, one-way ANOVA with respect to controls) (Fig. 8C–F. These findings confirm that α-syn exerts its effect by interfering with ω-MVIIC-sensitive Cav channels and D2-ARs. In a subset of experiments, current signals were sampled at 50 kHz to better quantify the $dV/dt$ values in the phase-plane plot analysis (Fig. 8G and H. We found

that even at this higher sampling rate, α-syn significantly increased $dV/dt_{max}$, while addition of MVIIC and sulp in α-syn-treated cells restored $dV/dt_{max}$ to control values: 75.34 ± 17.76 mV/ms in ctrl, 143.40 ± 40.22 mV/ms with α-syn ($P = 1.09E-4$), 68.12 ± 23.47 mV/ms with ω-MVIIC ($P = 0.0012***$) and 75.54 ± 27.71 mV/ms with sulp ($P = 0.0011$).

## Discussion

In this work, we quantified the effect of exogenous α-syn on AP firing and DA release in midbrain DA neurons dissociated from SN. As previously assessed by MEA recordings (Tomagra et al., 2023a), 48 h of incubation of α-syn (1 μM) in the culture medium induces the formation of oligomeric species that slows down the spontaneous firing activity, impairs burst generation and causes desynchronization of cultured midbrain neurons. Although MEAs are useful for comparing the effect of different doses of α-syn at progressive developmental stages of the same neuronal network, α-syn-specific action on DA neurons is hardly predictable in this configuration because of the involvement of glutamatergic and GABAergic inputs in the regulation of overall network activity.

Since, under physiological conditions, the firing activity of SN DA neurons and DA release itself is finely tuned by DA released through the D2-AR-mediated activation of GIRK2 channels (Anzalone et al., 2012; Dragicevic et al., 2014; Gantz et al., 2015; Hikima et al., 2021; Lüscher & Slesinger, 2010), the aim of the present work was to determine to what extent exogenous α-syn could alter the strict interplay between cell firing and DA release in SN DA neurons. To resolve this issue, we took advantage of patch-clamp recordings combined with μG-D-MEAs (Tomagra et al., 2019b) to respectively measure spontaneous firing, Cav-dependent secretion and real-time quantal DA release in SN DA neurons.

In our experimental conditions, exogenously applied α-syn drastically reduces the spontaneous firing rate of SN DA neurons, in good agreement with recent data (Hill et al., 2021) (Fig. 2), while significantly potentiating Cav2.2 channels (Fig. 7) and DA release (Figs 5 and 6). Since acute exposure to exogenous α-syn selectively potentiates Cav2.2 channels in cultured cortical neurons and upregulates DA release in striatal slices (Ronzitti et al., 2014), we hypothesized that the same might occur in SN DA neurons and that the potentiated DA release by α-syn could account for the reduction in spontaneous firing activity, through the D2-AR-mediated inhibitory pathway.

This hypothesis has been confirmed by three main findings. First, we show that the treatment with α-syn causes a selective potentiation of Cav2.2 channels.

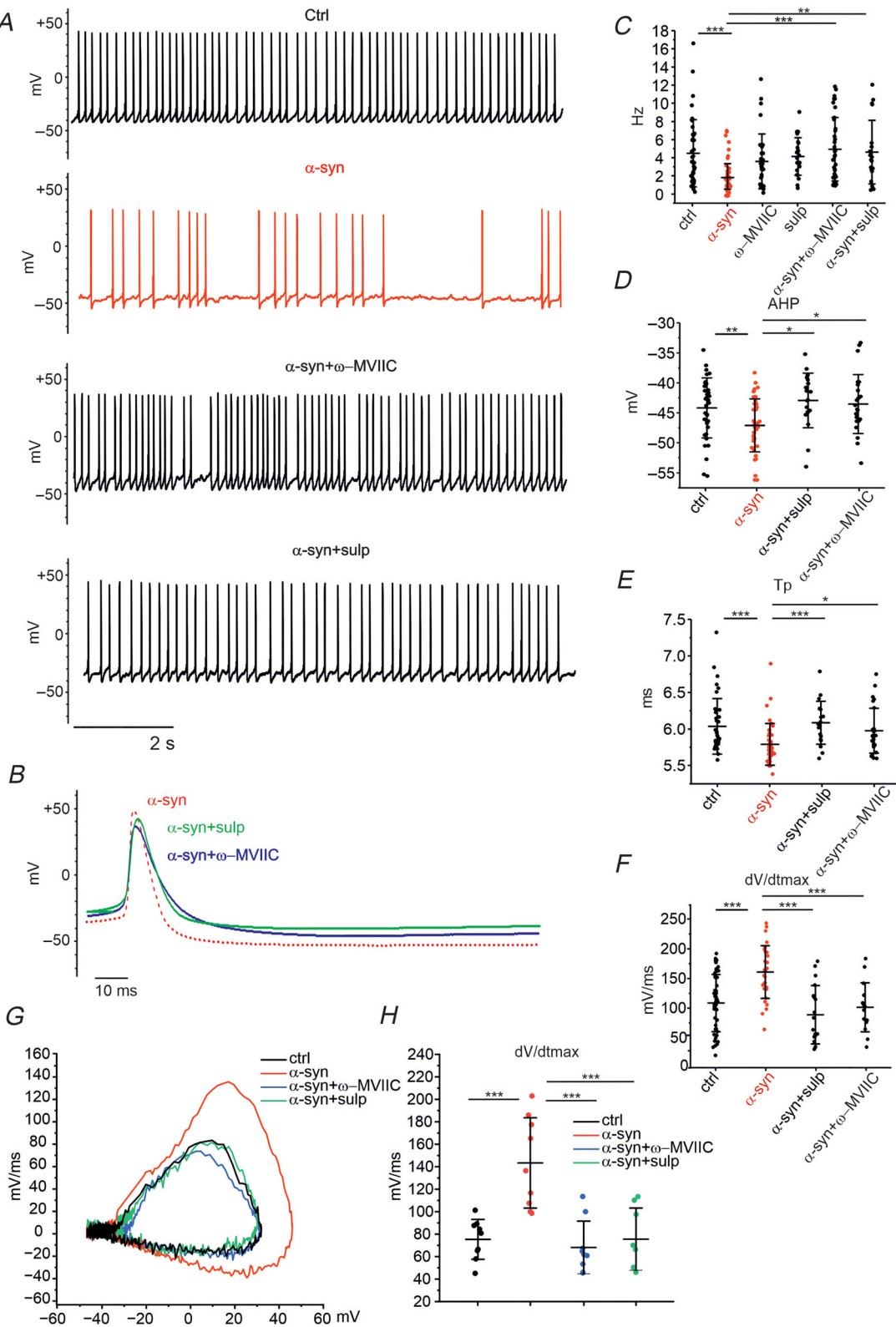

**Figure 8. The effects caused by α-syn on the firing activity and action potential waveform are restored by ω-MVIIC and/or sulpiride**

*A*, representative traces of spontaneous firing in control condition (ctrl), after α-syn incubation and after α-syn incubation plus ω-MVIIC and α-syn incubation plus sulpiride. *B*, a representative AP is shown at higher magnification. *C*, scatter plot of spontaneous firing rates. Data referred to ctrl (untreated neurons) and α-syn

($\alpha$-syn-treated neurons are reported from Fig. 2). *D*, scatter plot of after-hyperpolarization (AHP) values. *E*, AP time to peak. *F*, maximum time derivative voltage (d$V$/d$t_{max}$) calculated from traces sampled at 10 kHz, in control conditions, after $\alpha$-syn incubation, and after $\alpha$-syn incubation plus sulpiride and $\omega$-MVIIC. Statistically significant differences between ctrl and $\alpha$-syn, $\alpha$-syn and $\alpha$-syn + $\omega$-MVIIC, and $\alpha$-syn and $\alpha$-syn + sulp values are detailed in the text. *G*, phase-plane plot of AP calculated from traces sampled at 50 kHz. *H*, scatter plot of maximum time derivative voltage (d$V$/d$t_{max}$) in control conditions (black), after $\alpha$-syn incubation (red), and after $\alpha$-syn incubation plus $\omega$-MVIIC (blue) and plus sulpiride (green). $\alpha$-syn, $\alpha$-synuclein; AP, action potential.

The current carried by the latter has been measured in voltage-clamp experiments, by applying depolarizing steps at 0 mV, which corresponds to the potential at which Cav2.2 channels are effectively recruited during the steep depolarizing phase of spontaneous APs (Gantz et al., 2015). Through phase-plane plot analysis (Fig. 3), we effectively found that the slope of the AP rising phase is drastically increased by $\alpha$-syn while it is restored to the control value by the addition of $\omega$-MVIIC in $\alpha$-syn-treated cells. Second, the potentiated Cav2.2 channels cause an augmented Ca$^{2+}$-dependent secretion assayed by both capacitance change measurements (Fig. 5) and amperometric recordings using µG-D-MEA (Fig. 6). This result is in very good agreement with previous findings demonstrating the role of Cav2.2 channels in sustaining DA release in the basal ganglia (Bergquist et al., 1998). Our data show very clearly that the spontaneous quantal release of DA occurs at very low frequency in control conditions, while the frequency of quantal exocytotic events is drastically potentiated by $\alpha$-syn (Fig. 6). The low frequency of amperometric spikes detected under control matches with the one measured from cultured rat SN DA neurons using carbon fibre microelectrodes (Kim et al., 2008). In our experimental model, addition of $\omega$-MVIIC in $\alpha$-syn-treated neurons restores the normal frequency of quantal exocytotic events. Third, $\alpha$-syn causes a drastic inhibition of the spontaneous firing activity of DA neurons that is restored by adding $\omega$-MVIIC to the bath. This suggests a key role of $\omega$-MVIIC-sensitive Cav channels on the $\alpha$-syn-induced effects on AP firing. Our findings also show that $\omega$-MVIIC does not significantly affect the spontaneous firing rate of DA neurons, in good agreement with previous findings in which the N-type calcium channel blocker $\omega$-GVIA has been shown not to affect the spontaneous firing of midbrain DA neurons (Puopolo et al., 2007). It is also of note that while $\alpha$-syn reduces the basal firing rate, it has opposite effects when AP firing is induced by steps of increasing current amplitude (Fig. 2*F*). This is in good agreement with the observation that high-frequency AP firings counteract the auto-inhibitory effect of D2-AR via the facilitation of voltage-gated Cav channels, as recently reported (Sun et al., 2025).

After demonstrating that exogenous $\alpha$-syn potentiates Cav2.2-mediated DA release, our final goal was to test whether this mechanism could drive the D2-AR autocrine loop in SN DA neurons responsible for the decreases of AP firing frequency. Indeed, we found that administration

of the D2-antagonist sulp in $\alpha$-syn-treated cells restored both the spontaneous firing rate of the control condition as well as the AP shape (Fig. 8). Note that sulp *per se* has no significant effect on the basal firing rate in control conditions. This could be due to the limited spontaneous DA release in control conditions and also to the existence of different regulations of SN DA neurons by the released DA (Berretta et al., 2010).

As a final consideration, reported evidence showed a reduction of DA release by $\alpha$-syn during neurodegeneration (Decressac et al., 2012; Lundblad et al., 2012; Nuber et al., 2013). These findings are not necessarily in contrast with our results if we consider that the effect of misfolded $\alpha$-syn may depend on the age of the neuronal network, as shown for the reduced striatal DA release, observed in adult but not in young mice (Sun et al., 2022). Also, as reported in different models, increased DA release in early synaptic dysfunction may be a condition that precedes death of DA neurons (Goldberg et al., 2003; Lam et al., 2011; Wu et al., 2010).

Note that 2 day exposure of cultured neurons to $\alpha$-syn may trigger the activation of homeostatic plasticity mechanisms. Such mechanisms serve as compensatory processes that enable neurons to stabilize synaptic efficacy and network excitability in response to sustained perturbations. $\alpha$-Syn accumulation is known to interfere with synaptic transmission and to disturb the excitatory/inhibitory balance; consequently, neurons may engage long-term homeostatic adjustments, including the modulation of receptor expression, ion channel conductance or synaptic strength, to restore functional stability (Marino et al., 2022; Sharma & Burré, 2023).

In conclusion, our data highlight an early effect induced by exogenous $\alpha$-syn oligomers on SN DA neuronal activity that leads to increased Ca$^{2+}$ influx and potentiated DA release under basal conditions. As both events can indeed play a role in the initial stages of degeneration of SN DA neurons (Bisaglia et al., 2010), this mechanism could represent a target for future studies on rescue strategies to treat PD.

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

## Additional information

### Data availability statement

The data that support the findings of this study are available from the authors on reasonable request.

### Competing interests

All authors declare no competing interests.

### Author contributions

G.T. and A.B. designed, performed and analsed experiments; S.B., F.P., G.C. and A.d.I. contributed to design, and performed and analysed experiments; P.C., B.P., E.C., S.D.M., A.M. and V.C. critically revised the manuscript; C.F. prepared primary cultured neurons; V.C. designed the experiments and wrote the manuscript with input from all co-authors. All authors have read and approved the final version of the manuscript and agree to be accountable for all aspects of the work in ensuring that questions related to the accuracy or integrity of any part of the work are appropriately investigated and resolved. All persons designated as authors qualify for authorship, and all those who qualify for authorship are listed.

### Funding

Work by V. Carabelli, A. Marcantoni, F. Picollo and A. de Iure was supported by #NEXTGENERATIONEU (NGEU) and funded by the Ministry of University and Research (MUR), National Recovery and Resilience Plan (NRRP), project MNESYS (PE0000006) – A Multiscale integrated approach to the study of the Nervous System in Health and Disease (DN. 1553 11.10.2022), and by the Italian Ministry of Health, Ricerca Corrente (B. Picconi). PRIN PNRR MUR2022 (Prot. P2022CRAXJ) to A. Marcantoni. Progetto Trapezio (Compagnia di San Paolo) to V. Carabelli.

### Acknowledgements

Open access publishing facilitated by Universita degli Studi di Torino, as part of the Wiley - CRUI-CARE agreement.

### Keywords

alpha-synuclein, Cav2.2. channels, diamond-based micro-electrode arrays, quantal dopamine release, spontaneous firing

## Supporting information

Additional supporting information can be found online in the Supporting Information section at the end of the HTML view of the article. Supporting information files available:

**Peer Review History**
**Statistical Summary**

