## [Peer Review History · The Journal of Physiology]

α -synuclein oligomers slow-down action potential firing and enhance dopamine release by increasing Cav2.2 currents in midbrain dopaminergic neurons

Giulia Tomagra, Anthony Battaglia, Claudio Franchino, Sara Bonzano, Federico Piccolo, Giuseppe Chiantia, Antonio de Lure, Paolo Calabresi, Barbara Picconi, Emilio Carbone, Silvia De Marchis, Andrea Marcantoni, and Valentina Carabelli
DOI: 10.1113/JP288914

Corresponding author(s): Giulia Tomagra (giulia.tomagra@unito.it)

The following individual(s) involved in review of this submission have agreed to reveal their identity: zoe McElligot (Referee #1)

Review Timeline:

Submission Date:	19-Mar-2025
Editorial Decision:	12-May-2025
Revision Received:	05-Dec-2025
Editorial Decision:	23-Jan-2026
Revision Received:	04-Feb-2026
Accepted:	20-Feb-2026

Senior Editor: Katalin Toth

Reviewing Editor: Samuel Young

Transaction Report:

Dear Dr Tomagra,

Re: JP-RP-2025-288914 " **α -synuclein oligomers slow-down action potential firing and enhance dopamine release by increasing Cav2.2 currents in midbrain dopaminergic neurons**" by Giulia Tomagra, Anthony Battaglia, Claudio Franchino, Sara Bonzano, Federico Picollo, Silvia De Marchis, Antonio Delure, Paolo Calabresi, Barbara Picconi, Emilio Carbone, Andrea Marcantoni, and Valentina Carabelli

Thank you for submitting your manuscript to The Journal of Physiology. It has been assessed by a Reviewing Editor and by 2 expert referees and we are pleased to tell you that it is potentially acceptable for publication following satisfactory major revision.

LANGUAGE EDITING AND SUPPORT FOR PUBLICATION: If you would like help with English language editing, or other article preparation support, Wiley Editing Services offers expert help, including English Language Editing, as well as translation, manuscript formatting, and figure formatting at www.wileyauthors.com/eoo/preparation. You can also find resources for Preparing Your Article for general guidance about writing and preparing your manuscript at www.wileyauthors.com/eoo/prepresources.

REVISION CHECKLIST:

We look forward to receiving your revised submission.

Yours sincerely,

Katalin Toth
Senior Editor
The Journal of Physiology

REQUIRED ITEMS

- Include a Key Points list in the article itself, before the Abstract.

- Author photo and profile. First or joint first authors are asked to provide a short biography (no more than 100 words for one author or 150 words in total for joint first authors) and a portrait photograph. These should be uploaded and clearly labelled together in a Word document with the revised version of the manuscript. See Information for Authors for further details.

- The contact information for the person responsible for 'Research Governance' at your institution needs to be provided. This includes their name and an institutional email address. Please ensure the contact is not an author on this paper and provide an alternate contact if necessary, or confirm in the submission form that the author whose email was provided has sole responsibility for research governance. This is the person who is responsible for regulations, principles and standards of good practice in research carried out at the institution, for instance the ethical treatment of animals, the keeping of proper experimental records or the reporting of results.

- You must start the Methods section with a paragraph headed Ethical approval (https://jp.msubmit.net/cgi-bin/main.plex?form_type=display_requirements#methods).

Research must comply with The Journal's policies regarding animal experiments (<https://physoc.onlinelibrary.wiley.com/hub/animal-experiments>) and adherence to these policies must be stated in the manuscript.

Authors should confirm in their Methods section that their experiments were carried out according to the guidelines laid down by their institution's animal welfare committee, including an ethics approval reference number. The Methods section must contain a statement about access to food, water and housing, details of the anaesthetic regime: anaesthetic used, dose and route of administration, and method of killing the experimental animals.

- Your manuscript must include a complete Additional Information section, including competing interests; funding; author contributions and acknowledgements.

- Please upload separate high-quality figure files via the submission form.

- You must upload original, uncropped western blot/gel images (including controls) if they are not included in the manuscript. This is to confirm that no inappropriate, unethical or misleading image manipulation has occurred. These should be uploaded as 'Supporting information for review process only'. Please label/highlight the original gels so that we can clearly see which sections/lanes have been used in the manuscript figures. For more information, see: <https://physoc.onlinelibrary.wiley.com/hub/journal-policies#imagmanip>.

- Papers must comply with the Statistics Policy: https://jp.msubmit.net/cgi-bin/main.plex?form_type=display_requirements#statistics.

In summary:

- If $n \leq 30$, all data points must be plotted in the figure in a way that reveals their range and distribution. A bar graph with data points overlaid, a box and whisker plot or a violin plot (preferably with data points included) are acceptable formats.

- If $n > 30$, then the entire raw dataset must be made available either as supporting information, or hosted on a not-for-profit repository, e.g. FigShare, with access details provided in the manuscript.

- 'n' clearly defined (e.g. x cells from y slices in z animals) in the Methods. Authors should be mindful of pseudoreplication.

- All relevant 'n' values must be clearly stated in the main text, figures and tables.

- The most appropriate summary statistic (e.g. mean or median and standard deviation) must be used. Standard Error of the Mean (SEM) alone is not permitted.

- Exact p values must be stated. Authors must not use 'greater than' or 'less than'. Exact p values must be stated to three significant figures even when 'no statistical significance' is claimed.

- A Data Availability Statement is required for all papers reporting original data. This must be in the Additional Information section of the manuscript itself. It must have the paragraph heading 'Data Availability Statement'. All data supporting the results in the paper must be either: in the paper itself; uploaded as Supporting Information for Online Publication; or archived in an appropriate public repository. The statement needs to describe the availability or the absence of shared data. Authors must include in their statement: a link to the repository they have used, or a statement that it is available as Supporting Information; reference the data in the appropriate section(s) of their manuscript; and cite the data they have shared in the References section. Whenever possible, the scripts and other artefacts used to generate the analyses presented in the paper should also be publicly archived. If sharing data compromises ethical standards or legal requirements then authors are not expected to share it, but must note this in their statement. For more information, see our Statistics Policy.

- Please include an Abstract Figure file, as well as the Figure Legend text within the main article file. The Abstract Figure is a piece of artwork designed to give readers an immediate understanding of the research and should summarise the main conclusions. If possible, the image should be easily 'readable' from left to right or top to bottom. It should show the physiological relevance of the manuscript so readers can assess the importance and content of its findings. Abstract Figures should not merely recapitulate other figures in the manuscript. Please try to keep the diagram as simple as possible and without superfluous information that may distract from the main conclusion(s). Abstract Figures must be provided by authors no later than the revised manuscript stage and should be uploaded as a separate file during online submission labelled as File Type 'Abstract Figure'. Please also ensure that you include the figure legend in the main article file. All Abstract Figures should be created using BioRender. Authors should use The Journal's premium BioRender account to export high-resolution images. Details on how to use and access the premium account are included as part of this email.

- The corresponding author must provide an institutional email address (not a personal address) for their author account. We encourage ALL co-authors to also provide institutional email addresses. If this cannot be provided (as corresponding author), then a stamped letter must be provided from the institution which confirms their role and employment there (please upload this with the revised submission).

EDITOR COMMENTS

Reviewing Editor:

Both reviewers find this study to be impactful, interesting, makes conceptually advances, with the data largely supporting the conclusions. However, both reviewers have concerns. Both reviewers have concern that some of the conclusions based on the current data are not well supported. Reviewers had concern with respect to sampling rates used to acquire data for phase plot of analysis. Both point out that the sampling rates are too low. There is a lack of presynaptic calcium imaging which limits the ability to draw conclusions on synaptic calcium currents. Therefore, the authors will need to perform additional experiments in response to these two major concerns. There is a lack of detail in the methods section with respect to culture procedures and this will need to be added. Please list more detail on the embryo removal procedures. There is no information just listed references which is not so helpful. References to prior work is not sufficient. The authors will need to make text revisions to the manuscript in response to the positive comments and critiques. Finally, the manuscript will need to comply with Journal policies for reporting statistics as SD. Statistics need to be reported as SD. Currently it is reported as SEM.

REFeree COMMENTS

Referee #1:

In this manuscript Tomagra et al. demonstrate that alpha synuclein oligomers inhibit the firing rate of SN DA neurons, but also upregulates N-type Ca^{2+} currents and alters DA release. The authors use a culture model of midbrain DA neurons dissected from E16 TH-GFP mice, patch-clamp electrophysiology and amperometric recordings. The manuscript is interesting and uses the aforementioned techniques in conjunction with pharmacology to parse out the role of calcium channels and autocrine regulation of DA neuron firing. While I am highly enthusiastic, there are some things that should be considered prior to publication.

Major:

The sex of the mice in the culture condition needs to be reported. If both sexes were used were there any sex differences noted?

The authors need to report the temperature at which the recordings were made in the methods.

The authors may want to check and make sure that both the voltage clamp and current clamp recordings were filtered with a 1 kHz lowpass Bessel; the reviewer is not aware that there is a current clamp filter in EPC 10 HEKA amps.

In figure 2 there are no asterisks on panels C & D that would be helpful for the reader

It would be informative for the reader to understand the effect size using a Cohen's d or Hedges's g test as needed.

It is unclear if sampling at 10 kHz is going to resolve the action potential dynamics that the investigators are looking at, e.g. T_p is $5.8 \pm 0.04 - 6.0 \pm 0.9$ ms ...the standard error cannot be lower than the sampling rate. There are also issues with the phase-plot analysis at this sampling rate. The authors should record at 50 kHz sampling to demonstrate more resolution on these measurements. The lack of axon initial segment component (AIS bump, see Kevin Bender's work) looks to be missing in the control animals in fig 3B.

While I understand the reasoning to increase the calcium to amplify calcium currents and quantal release, this seems excessive to raise it to 10 mM! I would be interested to see if the parameters change in a more physiological concentration of Ca.

For figure 5, I would change "control" to baseline since that is a within prep experiment.

Figure 7 is labeled "Figure 4" in the legends in the PDF which rendered confusion.

While all of this data is interesting, it leaves one wondering what would happen in an aged brain with incubation with alpha-syn. Could these cells be investigated after an injection of the oligomers into the brain? An experiment like this would elevate the paper

Minor:

I would be consistent in calling channels either Cav1 or L-type, switching between them is difficult for novice readers.

Referee #2:

In this manuscript, the authors propose that exposure of cultured dopaminergic neurons from the substantia nigra to exogenous (extracellular) alpha-synuclein, induces changes in their electrophysiological properties. These changes result in a reduction in the frequency of spontaneously generated action potentials but an increase in the frequency of evoked APs. The halfwidth of the APs is narrower, time to peak shorter, after-hyperpolarization deeper, and their upstroke rate faster. Furthermore, by measuring depolarization-induced changes in capacitance while blocking sub-types of calcium channels, the authors show that neurons exposed to alpha-synuclein show a higher capacitance increase when CaV2.1/2.2 function, but not when CaV1/2.3 function. Moreover, alpha-synuclein exposure enhanced dopamine release events as measured using μ G-D-MEA amperometry, an effect that was annulled by blockage of CaV2.1/2.2 channels. CaV2.2 currents were shown to be enhanced by alpha-synuclein. Finally, blockage of D2 receptors or of CaV2.1/2.2 channels could rescue the alpha-synuclein-induced reduction in AP firing rates.

The authors propose that alpha-synuclein upregulates the activity of CaV2.2 channels in dopaminergic neurons, and that the enhanced release of dopamine inhibits spontaneous activity in the neurons in a feed-back mechanism.

While this report is interesting and of importance, it is my opinion that the authors did not address or explore several possible alternative explanations to their findings. Furthermore, it appears to me that some of the execution of some of the electrophysiological techniques was lacking.

Methodological

1. As far as I know, capacitance measurements are reliable when the cells are compact and close to being isopotential, due to the limited reach of voltage clamping across cellular segments in which the length constant (and the time constant) are not insignificant. The dopaminergic neurons in the cultures are not close to being isopotential, and the proposed change in capacitance conceptually mostly occurs in the synapses, which are electrically distant from the path site. Therefore, I am unsure about the reliability of the results presented in figure 4. A similar concern exists for the capability to perform reliable voltage clamp of synaptic currents to measure calcium transients (Fig. 6) which are conceptually mostly synaptic in nature (as in the case of P/Q VGCCs, for which a synaptic localization is assumed).

2. The sampling frequency does not appear to suffice for the generation of phase-plane plots in figure 3. A higher frequency would have helped to measure the dV/dt more reliably.

3. It appears to me that more information could have been extracted from the AP waveforms in figure 3. For example, is it possible that the threshold for AP generation differs? Is the slowly depolarizing phase prior to the upstroke of the AP due to a persistent sodium current that does not appear in the neurons exposed to alpha-synuclein?

4. Are the calcium currents in figure 4A different? Are these the same as addressed in figure 6? Why not refer to them here as well?

5. Presynaptic calcium imaging could have provided here significant value.

Interpretation

1. The authors did not discuss the possibility of homeostatic plasticity mechanisms. Since the cultured neurons were exposed to alpha-synuclein for two days, the activation of long-term compensatory plasticity is possible and should be considered.
2. Did the authors plan to address the question of GIRK channels experimentally? This would help in cementing the authors' interpretations.
3. The possible effect of alpha-synuclein on other neuronal cell types in regards to the results presented here should be discussed in more detail than that provided in the current manuscript (l. 515-518)
4. The authors show that sulpiride normalizes the effect of alpha-synuclein on AP frequency in figure 7. I found it difficult to understand how sulpiride did not affect the AP frequency under control conditions, considering that dopamine release presumably occurs under these conditions as well. Is the release of dopamine so low? In the case of spontaneous APs in dopaminergic neurons, the secretion of dopamine is not considered spontaneous (referring to figure 5 - in which I assume TTX was not used).

Minor

1. The description of the surgical removal of embryos was not described.
2. The authors refer to up-regulation of calcium channels. This term can be confusing because it can refer both to alteration of expression levels or to the functional properties of the channels themselves. The authors should be more direct in what is exactly meant. Did they mean up-regulation of calcium currents?
3. For figure 2E, it would be beneficial to show exemplary traces.
4. For figure 4: Did the authors perform measurements of currents when no calcium channels were blocked? Did these traces also differ for control and alpha-synuclein cells? For example, the authors reported comparison of results also without blocking calcium channels in figure 5.
5. The manuscript contains some typos and grammar mistakes. For example, "interpheres" in line 498, "a reduction" appears twice in l. 568. "Support" -> supporting (l. 55), etc.
6. The citation Sun et al (l. 559) is not complete.
7. Figure 7 is mislabeled as figure 4.

To conclude, in my opinion the manuscript is interesting and present important results, but should be improved both in terms of its interpretation and its methodology before being published.

END OF COMMENTS

EDITOR

COMMENTS

Reviewing

Editor:

Both reviewers find this study to be impactful, interesting, makes conceptually advances, with the data largely supporting the conclusions. However, both reviewers have concerns. Both reviewers have concern that some of the conclusions based on the current data are not well supported. Reviewers had concern with respect to sampling rates used to acquire data for phase plot of analysis. Both point out that the sampling rates are too low. There is a lack of presynaptic calcium imaging which limits the ability to draw conclusions on synaptic calcium currents. Therefore, the authors will need to perform additional experiments in response to these two major concerns. There is a lack of detail in the methods section with respect to culture procedures and this will need to be added. Please list more detail on the embryo removal procedures. There is no information just listed references which is not so helpful. References to prior work is not sufficient. The authors will need to make text revisions to the manuscript in response to the positive comments and critiques. Finally, the manuscript will need to comply with Journal policies for reporting **statistics as SD. Statistics need to be reported as SD.** Currently it is reported as SEM.

We thank the Editor and both Reviewers for their positive comments and suggestions that helped to improve the manuscript.

REFEREE COMMENTS

Referee #1:

In this manuscript Tomagra et al. demonstrate that alpha synuclein oligomers inhibit the firing rate of SN DA neurons, but also upregulates N-type Ca²⁺ currents and alters DA release. The authors use a culture model of midbrain DA neurons dissected from E16 TH-GFP mice, patch-clamp electrophysiology and amperometric recordings. The manuscript is interesting and uses the aforementioned techniques in conjunction with pharmacology to parse out the role of calcium channels and autocrine regulation of DA neuron firing. While I am highly enthusiastic, there are somethings that should be considered prior to publication.

Major:

The sex of the mice in the culture condition needs to be reported. If both sexes were used were there any sex differences noted?

The culture was obtained from mouse embryos, so sex cannot be distinguished.

The authors need to report the temperature at which the recordings were made in the methods.

We thank the Reviewer for requesting this clarification. All experiments have been performed at room temperature. Both patch-clamp and diamond micro-electrode arrays recordings lasted no longer than 30 minutes. We have included this sentence in the text.

The authors may want to check and make sure that both the voltage clamp and current clamp recordings were filtered with a 1 kHz lowpass Bessel; the reviewer is

not aware that there is a current clamp filter in EPC 10 HEKA amps.

We thank the reviewer for this observation. Yes, we confirm that both the voltage clamp and current clamp recordings were filtered with a 1 kHz lowpass Bessel. We used an EPC 10 USB. As reported in the HEKA manual, section 5.3 Amplifier Parameters :“ Filter: controls an analog 4-pole low-pass filter for current monitor 2 (Imon2) and the voltage monitor (Vmon). Dragging the mouse or typing allows fine adjustment from 0.1– 16 kHz in 0.1 kHz steps (guaranteed accuracy 0.5– 15 kHz)”

In figure 2 there are no asterisks on panels C & D that would be helpful for the reader

In order to meet the Reviewer’s request, we have added the asterisks in all figures, according to the following criterium: (=0.05, **=0.01, ***=0.001). Since it is requested by The Journal of Physiology to indicate the exact Pvalue, we have kept the exact Pvalues in the figure legends.*

It would be informative for the reader to understand the effect size using a Cohen's d or Hedges's g test as added.

We thank the reviewer for this suggestion, but in our view this test is not properly useful for our conditions.

As reported in [1-3], this test is applicable when comparing similar conditions drawn from different pools, especially when the statistical samples are normally distributed, have a low dispersion index, and have sample sizes greater than 20. Our data do not meet any of the above requirements.

[1] “In Calculating and reporting effect sizes to facilitate cumulative science: a practical primer for t-tests and ANOVAs” Daniël Lakens REVIEW article, *Front. Psychol.*, 26 November 2013 Sec. Cognition Volume 4 2013. <https://doi.org/10.3389/fpsyg.2013.00863>

[2] “Interpretation of the Standardized Mean Difference Effect Size When Distributions Are Not Normal or Homoscedastic” Larry V Hedges. *Educ Psychol Meas.* 2024 Oct 6;85(2):245–257. doi: 10.1177/00131644241278928

[3] “Statistical Power Analysis for the Behavioral Sciences”, Jacob Cohen, 2° Edition, L. Erlbaum Associates, 1988

It is unclear if sampling at 10 kHz is going to resolve the action potential dynamics that the investigators are looking at, e.g. T_p is $5.8 + 0.04 - 6.0 + 0.9$ msthe standard error cannot be lower than the sampling rate.

We thank the Reviewer for this comment. We absolutely agree that the standard deviation cannot be lower than the sampling rate. Though, in the original version of the manuscript, data have been reported with the mean standard error, instead of the standard deviation. In this revised version, as requested by the Journal, data have been indicated with the standard deviation values. For example, as reported in Results, page 7, T_p is 6.0 ± 0.4 ms in control and 5.8 ± 0.4 ms with α -syn. Thus the standard deviation is greater than the sampling rate.

Similar conclusions can be drawn also using a higher sampling frequency (50 KHz). In this case, T_p is 6.5 ± 0.3 ms for controls and 5.4 ± 0.3 ms for α -syn treated cells ($p=0.005$), suggesting the statistical difference among controls and treated cells is not changed.

There are also issues with the phase-plot analysis at this sampling rate. The authors should record at 50 kHz sampling to demonstrate more resolution on these measurements. The lack of axon initial segment component (AIS bump, see Kevin Bender's work) looks to be missing in the control animals in fig 3B.

We thank the reviewer for this accurate observation. As suggested, we performed a new set of experiments using 50 kHz sampling rate for measuring the phase-plane plot. A new figure (Fig.4 in the revised manuscript) has been added to this purpose. Statistical difference of dV/dt values among controls and treated cells still persists and main conclusions are not changed.

*For what concerns the lack of **axon initial segment component**, we found that the presence of the kink is independent of the sampling rate and the presence of α -synuclein, as shown in the new Fig.4 and detailed in the text.*

While I understand the reasoning to increase the calcium to amplify calcium currents and quantal release, this seems excessive to raise it to 10 mM! I would be interested to see if the parameters change in a more physiological concentration of Ca.

We agree with the Reviewer that using 2 mM Ca^{2+} represents indeed a more physiological condition. Nevertheless, we used 10 mM Ca^{2+} instead of 2mM Ca^{2+} because the capacitance trace is very noisy in 2 mM Ca^{2+} , and small capacitance increases (< 20 fF) may be masked by noise or automatic lock-in circuit correction. This choice should not affect the main conclusion, since we are comparing the secretory responses under control and after treatment with α -synuclein.

The use of external $CaCl_2$ in the range 5-10 mM has also been reported by Erwin Neher's group while studying capacitance changes in mouse chromaffin cells (see for instance Moser & Neher, Proc. Natl. Acad. Sci. USA Vol. 94, pp. 6735–6740, 1997).

For figure 5, I would change "control" to baseline since that is a within prep experiment.

We thank the Reviewer for this observation, we changed the term "control" to "baseline".

Figure 7 is labeled "Figure 4" in the legends in the PDF which rendered confusion.

The mislabeling has been corrected.

While all of this data is interesting, it leaves one wondering what would happen in an aged brain with incubation with alpha-syn. Could these cells be investigated after an injection of the oligomers into the brain? An experiment like this would elevate the paper

We absolutely agree with the Reviewer that investigating the effect in the aged brain would represent a relevant issue. In-vivo experiments performed by injecting oligomers into the brain represents the logical progression of this work. Though, this is a very different model, as in our case we are monitoring the effect of oligomers during network development, being the culture obtained from embryos. As such, we are planning future work in this direction, but this approach needs dedicated new research funds.

Minor:

I would be consistent in calling channels either Cav1 or L-type, switching between them is difficult for novice readers.
We agree with the Reviewer and we have standardized the nomenclature.

Referee#2:

In this manuscript, the authors propose that exposure of cultured dopaminergic neurons from the substantia nigra to exogenous (extracellular) alpha-synuclein, induces changes in their electrophysiological properties. These changes result in a reduction in the frequency of spontaneously generated action potentials but an increase in the frequency of evoked APs. The halfwidth of the APs is narrower, time to peak shorter, after-hyperpolarization deeper, and their upstroke rate faster. Furthermore, by measuring depolarization-induced changes in capacitance while blocking sub-types of calcium channels, the authors show that neurons exposed to alpha-synuclein show a higher capacitance increase when CaV2.1/2.2 function, but not when CaV1/2.3 function. Moreover, alpha-synuclein exposure enhanced dopamine release events as measured using μ G-D-MEA amperometry, an effect that was annulled by blockage of CaV2.1/2.2 channels. CaV2.2 currents were shown to be enhanced by alpha-synuclein. Finally, blockage of D2 receptors or of CaV2.1/2.2 channels could rescue the alpha-synuclein-induced reduction in AP firing rates.

The authors propose that alpha-synuclein upregulates the activity of CaV2.2 channels in dopaminergic neurons, and that the enhanced release of dopamine inhibits spontaneous activity in the neurons in a feed-back mechanism.

While this report is interesting and of importance, it is my opinion that the authors did not address or explore several possible alternative explanations to their findings. Furthermore, it appears to me that some of the execution of some of the electrophysiological techniques was lacking.

Methodological

1. As far as I know, capacitance measurements are reliable when the cells are compact and close to being isopotential, due to the limited reach of voltage clamping across cellular segments in which the length constant (and the time constant) are not insignificant. The dopaminergic neurons in the cultures are not close to being isopotential, and the proposed change in capacitance conceptually mostly occurs in the synapses, which are electrically distant from the path site. Therefore, I am unsure about the reliability of the results presented in figure 4. A similar concern exists for the capability to perform reliable voltage clamp of synaptic currents to measure calcium transients (Fig. 6) which are conceptually mostly synaptic in nature (as in the case of P/Q VGCCs, for which a synaptic localization is assumed).

We agree with the Reviewer and we are aware of the fact that the reliability of capacitance recordings require isopotential recording conditions. Though, it is worth noticing that in SN DA neurons both axonal and somatodendritic dopamine release occur and these processes have been extensively investigated by several groups (see for instance Chen et al., 2011, doi: 10.3389/fnsys.2011.00039), and interpreted as mainly ascribed to the somatic component. Detection of somatodendritic release using membrane capacitance

increases from neurons has been also performed by other groups, for example (de Kock et al, J. Neuroscience, 2003), or in other models experimental models (Groten, J. Neuroscience 2015; Hartveit et al., Physiological Reports 2019). Concerning the detection of P/Q channels at the somatic level in dopaminergic neurons, this has been already demonstrated by other groups (see Cardozo and Bean, 1995).

2. The sampling frequency does not appear to suffice for the generation of phase-plane plots in figure 3. A higher frequency would have helped to measure the dV/dt more reliably.

We thank the Reviewer for raising this issue. As suggested, we repeated the AP recordings by sampling at 50 kHz frequency. The new data related to the dV/dt values have now been included in a new figure (Fig. 4). These new data confirm that, also at 50 kHz sampling rate, α -synuclein significantly increases the maximum dV/dt value (from 75.34 ± 17.76 mV/ms in ctrl, to 143.40 ± 40.22 mV/ms in α -syn, $p=0.0011$).

We also performed new experiments for measuring dV/dt values (50 kHz sampling rate) in cells treated with α -synuclein and MVIIC or sulpiride. The new values have now been added in Fig.8 (panel H). Our data show that also using this higher sampling rate, dV/dt values for controls are not significantly different from those obtained in the presence of α -synuclein plus MVIIC or α -synuclein sulpiride, thus the main conclusion remains unaffected.

3. It appears to me that more information could have been extracted from the AP waveforms in figure 3. For example, is it possible that the threshold for AP generation differs?

We thank the Reviewer for raising this issue. Concerning the AP threshold, we observed that it was significantly decreased after α -syn incubation (from -39.4 ± 3.8 mV in control condition to -41.9 ± 4.5 mV with α -syn, $p=0.038$, Fig.3H). We added this additional result to the text and in figure 3, as suggested.

Is the slowly depolarizing phase prior to the upstroke of the AP due to a persistent sodium current that does not appear in the neurons exposed to alpha-synuclein?

We thank the Reviewer for this observation. Even though we cannot exclude a role for the persistent sodium current, it has been previously reported (Khaliq & Bean, The Journal of Neuroscience, 2010) that the background leak current is the dominant current driving pacemaking in VTA neurons, while the same current is present only in a fraction of SN neurons, and, when present, is generally much smaller than the one recorded in VTA (0.19 pA/pF versus 1.35 pA/pF in VTA).

4. Are the calcium currents in figure 4A different? Are these the same as addressed in figure 6? Why not refer to them here as well?

Calcium currents reported in Fig. 4 and 6 derive from two different set of experiments.

Fig. 4 (now renamed Fig 5), focuses on the measurement of capacitance increase.

On the contrary, in Fig. 6 (now renamed Fig. 7), the specific effect of α -syn on different calcium channels (Cav1, Cav2.1 and Cav2.2) has been reported, to demonstrate that α -syn selectively increases Ca_v2.2 currents.

5. Presynaptic calcium imaging could have provided here significant value.

We agree with the Reviewer on the fact that presynaptic calcium imaging could provide significant new insights. It is worth noticing that, to measure presynaptic calcium imaging, the use of the genetically encoded Ca²⁺ indicator SyGCaMP6s is currently a challenging

approach, as it requires neuronal infection. This is the reason why we used alternative approaches such as capacitance measurements, voltage-clamp and amperometry. Indeed, the point raised by the Reviewer opens a very interesting issue related to the possible role that increased Ca^{2+} may exert on intracellular Ca^{2+} signaling. This topic is worth to be investigated in the future and requires new sets experiments that we are addressing in an incoming new research project.

Interpretation

1. The authors did not discuss the possibility of homeostatic plasticity mechanisms. Since the cultured neurons were exposed to alpha-synuclein for two days, the activation of long-term compensatory plasticity is possible and should be considered.

We thank the Reviewer for this observation. We added a sentence in the Discussion paragraph. "It is worth mentioning that two-day exposure of cultured neurons to alpha-synuclein may trigger the activation of homeostatic plasticity mechanisms. Such mechanisms serve as compensatory processes that enable neurons to stabilize synaptic efficacy and network excitability in response to sustained perturbations. Alpha-synuclein accumulation is known to interfere with synaptic transmission and to disturb the excitatory/inhibitory balance; consequently, neurons may engage long-term homeostatic adjustments, including the modulation of receptor expression, ion channel conductance, or synaptic strength, to restore functional stability". This certainly represents an excellent issue for future investigations.

2. Did the authors plan to address the question of GIRK channels experimentally? This would help in cementing the authors' interpretations.

We thank the Reviewer for this important consideration. This is a very hot topic in research regarding the effects of α -synuclein on DA neurons, as GIRK activity is part of the inhibitory feedback mediated by D2 autoreceptors. Numerous studies [1-3] focused on this pathway, but due to its complexity, it deserves a dedicated study. In this paper, our focus was on the effect of Cav channels, specifically on the modulation of Cav 2.2. Our next step will be to add this piece of information and explore the possible relationship between the two phenomena, which could be correlated, or rather, two different causes.

1. "In Parkinson's patient-derived dopamine neurons, the triplication of α -synuclein locus induces distinctive firing pattern by impeding D2 receptor autoinhibition" (Lin et al., 2021)
2. "Calbindin and *Girk2/Aldh1a1* define resilient vs vulnerable dopaminergic neurons" (del Rey et al., 2024)
3. "Potassium Channels in Parkinson's Disease: Potential Roles ..." (Chen et al., 2023)

3. The possible effect of alpha-synuclein on other neuronal cell types in regards to the results presented here should be discussed in more detail than that provided in the current manuscript (l. 515-518).

We thank the reviewer for this observation. Because the literature on the effects of α -synuclein is impressively huge, since it includes different animal models, variable forms of aggregated α -syn (oligomers, PFF, overexpressed α -syn), as well as different experimental models (for examples cultured cell versus slices), we have limited our discussion to the effect of exogenous α -syn on cultured neurons.

Nonetheless, in this revised version we added new references focused on the effects induced by exogenous α -synuclein

4. The authors show that sulpiride normalizes the effect of alpha-synuclein on AP frequency in figure 7. I found it difficult to understand how sulpiride did not affect the AP frequency under control conditions, DA considering that dopamine release presumably occurs under these conditions as well. Is the release of dopamine so low? In the case of spontaneous APs in dopaminergic neurons, the secretion of dopamine is not considered spontaneous (referring to figure 5 - in which I assume TTX was not used).

We thank the Reviewer for giving the possibility of clarifying this issue. The Reviewer interpretation is correct, the basal DA release is very low. As we demonstrated in our previous work (Tomagra et al. 2019), basal release is 0.11 ± 0.07 Hz. For this reason, the effect of sulpiride is negligible in untreated cells.

The Reviewer is right, dopamine secretion cannot be considered spontaneous as TTX was not present in the extracellular solution.

Minor

1. The description of the surgical removal of embryos was not described.

A detailed paragraph has been added in Materials and Methods, paragraph "Primary cell culture of embryonic midbrain neurons".

2. The authors refer to up-regulation of calcium channels. This term can be confusing because it can refer both to alteration of expression levels or to the functional properties of the channels themselves. The authors should be more direct in what is exactly meant. Did they mean up-regulation of calcium currents?

We thank the Reviewer for raising this issue. Yes, indeed the used term was misleading. We have now substituted up-regulation of calcium channels with up-regulation of calcium currents.

3. For figure 2E, it would be beneficial to show exemplary traces.

As requested, representative traces have been added in Figure 2.

4. For figure 4: Did the authors perform measurements of currents when no calcium channels were blocked? Did these traces also differ for control and alpha-synuclein cells? For example, the authors reported comparison of results also without blocking calcium channels in figure 5.

We thank the reviewer for this observation. Experiments without channel blockers and related statistics have been included in figure 7 (panel H). A sentence has been added in the manuscript main text.

5. The manuscript contains some typos and grammar mistakes. For example, "interpheres" in line 498, "a reduction" appears twice in l. 568. "Support" -> supporting (l. 55), etc.

We thank the reviewer for these corrections; we have made the requested changes and double-checked the work.

6. The citation Sun et al (l. 559) is not complete.

The citation has been corrected.

7. Figure 7 is mislabeled as figure 4.

We have corrected the mislabeling.

To conclude, in my opinion the manuscript is interesting and present important results, but should be improved both in terms of its interpretation and its methodology before being published.

Dear Dr Tomagra,

Re: JP-RP-2025-288914R1 " **α -synuclein oligomers slow-down action potential firing and enhance dopamine release by increasing Cav2.2 currents in midbrain dopaminergic neurons**" by Giulia Tomagra, Anthony Battaglia, Claudio Franchino, Sara Bonzano, Federico Picollo, Giuseppe Chiantia, Antonio de Iure, Paolo Calabresi, Barbara Picconi, Emilio Carbone, Silvia De Marchis, Andrea Marcantoni, and Valentina Carabelli

Thank you for submitting your manuscript to The Journal of Physiology. It has been assessed by a Reviewing Editor and by 2 expert referees and we are pleased to tell you that it is acceptable for publication following satisfactory revision.

REVISION CHECKLIST:

Please upload two versions of your manuscript text: one with all relevant changes highlighted and one clean version with no changes tracked. The manuscript file should include all tables and figure legends, but each figure/graph should be uploaded as separate, high-resolution files. The journal is now integrated with Wiley's Image Checking service. For further details,

see: <https://www.wiley.com/en-us/network/publishing/research-publishing/trending-stories/upholding-image-integrity-wileys-image-screening-service>

We look forward to receiving your revised submission.

Yours sincerely,

Katalin Toth
Senior Editor
The Journal of Physiology

REQUIRED ITEMS

1) - Please include an Abstract Figure legend. An appropriate figure legend, which should not exceed 150 words in length, should be included in the main manuscript file.

2) - Please include a full title page as part of your main article (Word) file, which should contain the following: title, authors, affiliations, corresponding author name and contact details, keywords, and running title.

EDITOR COMMENTS

Reviewing Editor:

The authors have done a good job of responding to previous critiques. Please make text edits in response to reviewer#2.

REFEREE COMMENTS

Referee #1:

The authors have largely satisfied my concerns.

Referee #2:

I find the revised manuscript by Tomagra et al. to be substantially improved compared to the original version.

The topic of the research is of substantial interest in the field. It enhances our understanding of the possible mechanisms by which secreted alpha-synuclein may alter basic neuronal properties, synaptic transmission, and the effect of alpha-synuclein (and dopamine secretion) on the activity of dopaminergic neurons.

The research is well-planned and well-performed. The science presented by it exhibits the next and necessary step following the previously published data by the authors. It is not just an increment; it also supplies a good explanation and model to account for the previously reported results.

The conclusions of this study are warranted. The authors also address various questions and possibilities that are related to their interpretations.

I have a few minor comments to add:

1. The authors indicate that the AP overshoot value is not altered by asyn incubation in the 10KHz recordings. To my eye, in the 50 KHz recordings, it is possible that a difference is observed. However, only statistics can tell if this is the case.
2. When discussing the AP threshold, I don't think that the adjective to use is described the effect of asyn is that it was "decreased" but rather that it was "lowered".
3. In line 407 in the results, please indicate that the toxin is SNX-482 (as mentioned in the methods section), considering there are several SNX toxins.
4. If I understand correctly, in the experiments described by Figure 5, neither sodium nor potassium channel blockers were used. Therefore, more care should be taken when discussing the meaning of the measured quantity of charge.
5. When comparing the values in Figure 5C and F, am I to understand that most of the charge change under control conditions is due to CaV1 or 2.3 channels?
6. In Fig. 6, the authors show the effect of asyn incubation in the same cultures previously recorded in the μ G-D-MEA chambers. This configuration is strong because it allows paired comparisons (which I am unsure if they were used here). However, this means that in claiming that asyn increases the frequency and other parameters, the authors did not consider the contribution of the time that passed during the incubation (2 days). The required control for this is to record untreated wells 2 days apart, in parallel with the treated wells. I do not think the authors need to perform new experiments - they just need to explain why this is not an issue.
7. In relation to the experiments shown in Figure 8, why didn't the authors apply a D2-selective agonist to control neurons?
8. In line 645, "interphere" should be "interfere".

END OF COMMENTS

I find the revised manuscript by Tomagra et al. to be substantially improved compared to the original version.

The topic of the research is of substantial interest in the field. It enhances our understanding of the possible mechanisms by which secreted alpha-synuclein may alter basic neuronal properties, synaptic transmission, and the effect of alpha-synuclein (and dopamine secretion) on the activity of dopaminergic neurons.

The research is well-planned and well-performed. The science presented by it exhibits the next and necessary step following the previously published data by the authors. It is not just an increment; it also supplies a good explanation and model to account for the previously reported results.

The conclusions of this study are warranted. The authors also address various questions and possibilities that are related to their interpretations.

I have a few minor comments to add:

1. The authors indicate that the AP overshoot value is not altered by asyn incubation in the 10KHz recordings. To my eye, in the 50 KHz recordings, it is possible that a difference is observed. However, only statistics can tell if this is the case.

We thank the reviewer for the comment. We performed a statistical analysis of the overshoot values, and found that the data were not significantly different ($p = 0.11$, Mann Whitney test).

When discussing the AP threshold, I don't think that the adjective to use is described the effect of asyn is that it was "decreased" but rather that it was "lowered".

We thank the reviewer for this suggestion. We replaced the adjective, as suggested.

3. In line 407 in the results, please indicate that the toxin is SNX-482 (as mentioned in the methods section), considering there are several SNX toxins.

We thank the reviewer for noticing this inaccuracy, we have corrected the wording.

4. If I understand correctly, in the experiments described by Figure 5, neither sodium nor potassium channel blockers were used. Therefore, more care should be taken when discussing the meaning of the measured quantity of charge.

Potassium channel blockers are present in the extracellular solution (4 mM TEA-Cl). Tetrodotoxin (TTX) was not added to prevent slowdown of Na^+ -channel gating kinetics (see Horrigan and Bookman, 1994, Carabelli et al., 2003, Biophysical Journal, Volume 85, 1326–1337). The quantity of charge Q was calculated as the time integral of the inward current. Given the presence of an early inward Na^+ current [since TTX was not used], the limits for the current integration were fixed 3–4 ms after the beginning of the pulse once 80% of the Na^+ currents were decayed. We included this clarification in the text.

5. When comparing the values in Figure 5C and F, am I to understand that most of the charge change under control conditions is due to Cav1 or 2.3 channels?

Yes, under control conditions, the quantity of charge carried by Cav1 and Cav2.3 is higher than the one carried by Cav2.1 and Cav2.2.

6. In Fig. 6, the authors show the effect of asyn incubation in the same cultures previously recorded in the $\mu\text{G-D-MEA}$ chambers. This configuration is strong because it allows paired comparisons (which I am

unsure if they were used here). However, this means that in claiming that α -syn increases the frequency and other parameters, the authors did not consider the contribution of the time that passed during the incubation (2 days). The required control for this is to record untreated wells 2 days apart, in parallel with the treated wells. I do not think the authors need to perform new experiments - they just need to explain why this is not an issue.

We thank the reviewer for this concern.

Different experimental evidence demonstrates that at 14 DIV DA neurons are considered mature. In Tomagra et al., 2019 [1], we have already shown that cultures up to 20 DIV show a very low release rate and these values are in good agreement with other studies [2-3]. For these reasons we do not expect that exocytosis may significantly change among 11 and 14 DIV.

[1] Tomagra G, Picollo F, Battiato A, Picconi B, De Marchis S, Pasquarelli A, Olivero P, Marcantoni A, Calabresi P, Carbone E & Carabelli V. (2019b). Quantal Release of Dopamine and Action Potential Firing Detected in Midbrain Neurons by Multifunctional Diamond-Based Microarrays. *Frontiers in neuroscience* **13**, 288.

[2] Staal R, Mosharov E, Sulzer D, (2004). Dopamine neurons release transmitter via a flickering fusion pore. *Nature Neuroscience*. Volume 7, <https://doi.org/10.1038/nn1205>.

[3] Pothos E, Davila V, Sulzer D (1998). Presynaptic Recording of Quanta from Midbrain Dopamine Neurons and Modulation of the Quantal Size. *Journal of Neuroscience* 1 June 1998, 18 (11) 4106-4118; <https://doi.org/10.1523/JNEUROSCI.18-11-04106.1998>

7. In relation to the experiments shown in Figure 8, why didn't the authors apply a D2-selective agonist to control neurons?

We thank the reviewer for this observation. We decided to use the selective antagonist sulpiride to prove that D2 receptors are involved, as this hypothesis is confirmed by the "rescue" experiments performed with α -syn + sulpiride. Alternatively, the addition of a D2-selective agonist would eventually increase the inhibiting effect of α -syn on the firing rate but would not give additional information on a possible "rescue" mechanism.

8. In line 645, "interphere" should be "interfere".

We thank the reviewer and we have made the correction accordingly.

Dear Dr Tomagra,

Re: JP-RP-2026-288914R2 " **α -synuclein oligomers slow-down action potential firing and enhance dopamine release by increasing Cav2.2 currents in midbrain dopaminergic neurons**" by Giulia Tomagra, Anthony Battaglia, Claudio Franchino, Sara Bonzano, Federico Picollo, Giuseppe Chiantia, Antonio de Iure, Paolo Calabresi, Barbara Picconi, Emilio Carbone, Silvia De Marchis, Andrea Marcantoni, and Valentina Carabelli

We are pleased to tell you that your paper has been accepted for publication in The Journal of Physiology.

Yours sincerely,

Katalin Toth
Senior Editor
The Journal of Physiology

IMPORTANT POINTS TO NOTE FOLLOWING ACCEPTANCE OF YOUR PAPER:

- **IMPORTANT NOTICE ABOUT OPEN ACCESS:** To assist authors whose funding agencies mandate immediate public access to published research findings, The Journal of Physiology allows authors to pay an Open Access (OA) fee to have their papers made freely available immediately on publication.

The Corresponding Author will receive an email from Wiley with details on how to register or log in to Wiley Authors where you will be able to place an order.

- You can check if your funder or institution has a Wiley Open Access Account here:
<https://authors.wiley.com/author-resources/Journal-Authors/open-access/author-compliance-tool.html>

- You can help your research get the attention it deserves! Check out Wiley's free Promotion Guide for best-practice recommendations for promoting your work at: www.wileyauthors.com/eeo/guide. You can learn more about Wiley Editing Services which offers professional video, design, and writing services to create shareable video abstracts, infographics, conference posters, lay summaries, and research news stories for your research at: www.wileyauthors.com/eeo/promotion.

- If you would like to receive our 'Research Roundup', a monthly newsletter highlighting the cutting-edge research published in The Physiological Society's family of journals (The Journal of Physiology, Experimental Physiology, Physiological Reports, The Journal of Nutritional Physiology and The Journal of Precision Medicine: Health and Disease), please click this link, fill in your name and email address and select 'Research Roundup':
<https://www.physoc.org/journals-and-media/membernews>

EDITOR COMMENTS

Reviewing Editor:

The authors have done an excellent job responding to critiques. There are no further concerns.